# Identification of optimal dosing schedules of dacomitinib and osimertinib for a phase I/II trial in advanced *EGFR*-mutant non-small cell lung cancer

Kamrine E. Poels [1,2], Adam J. Schoenfeld[3], Alex Makhnin[3], Yosef Tobi [3], Yuli Wang[4], Heidie Frisco-Cabanos[5], Shaon Chakrabarti[1,2,6], Manli Shi[4], Chelsi Napoli[5], Thomas O. McDonald[1,2,6,7], Weiwei Tan[8], Aaron Hata[5,9,10], Scott L. Weinrich[4], Helena A. Yu[3]✉ & Franziska Michor [1,2,6,7,9,11]✉

Despite the clinical success of the third-generation EGFR inhibitor osimertinib as a first-line treatment of *EGFR*-mutant non-small cell lung cancer (NSCLC), resistance arises due to the acquisition of *EGFR* second-site mutations and other mechanisms, which necessitates alternative therapies. Dacomitinib, a pan-HER inhibitor, is approved for first-line treatment and results in different acquired *EGFR* mutations than osimertinib that mediate on-target resistance. A combination of osimertinib and dacomitinib could therefore induce more durable responses by preventing the emergence of resistance. Here we present an integrated computational modeling and experimental approach to identify an optimal dosing schedule for osimertinib and dacomitinib combination therapy. We developed a predictive model that encompasses tumor heterogeneity and inter-subject pharmacokinetic variability to predict tumor evolution under different dosing schedules, parameterized using in vitro dose-response data. This model was validated using cell line data and used to identify an optimal combination dosing schedule. Our schedule was subsequently confirmed tolerable in an ongoing dose-escalation phase I clinical trial (NCT03810807), with some dose modifications, demonstrating that our rational modeling approach can be used to identify appropriate dosing for combination therapy in the clinical setting.

[1] Department of Biostatistics, Harvard T.H. Chan School of Public Health, Boston, MA, USA. [2] Department of Data Science, Dana Farber Cancer Institute, Boston, MA, USA. [3] Division of Solid Tumor Oncology, Department of Medicine, Thoracic Oncology Service, Memorial Sloan-Kettering Cancer Center, Weill Cornell Medical College, New York, NY, USA. [4] Oncology Research and Development, Pfizer Inc, La Jolla, CA, USA. [5] Massachusetts General Hospital Cancer Center, Boston, MA, USA. [6] Department of Stem Cell and Regenerative Biology, Harvard University, Cambridge, MA, USA. [7] The Center for Cancer Evolution, Dana-Farber Cancer Institute, Boston, MA, USA. [8] Clinical Pharmacology Oncology, Global Product Development, Pfizer Inc, San Diego, CA, USA. [9] The Ludwig Center at Harvard, Boston, MA, USA. [10] Department of Medicine, Harvard Medical School, Boston, MA, USA. [11] The Broad Institute of MIT and Harvard, Cambridge, MA, USA. ✉email: yuh@mskcc.org; michor@jimmy.harvard.edu

Activating *EGFR* mutations are present in 15% of all non-small cell lung cancers and identify the subset of lung cancers that are sensitive to EGFR tyrosine kinase inhibitors (TKIs)[1]. Despite the fact that most patients with *EGFR*-mutant (*EGFR*-m) NSCLC have robust responses to EGFR TKIs, the majority ultimately develop disease progression. Osimertinib, a third-generation, irreversible EGFR TKI, has become a standard of care first-line treatment for patients with *EGFR*-mutant lung cancers. Osimertinib was initially approved to address acquired EGFR T790M, the most frequent mechanism of acquired resistance to earlier generation EGFR TKIs, and has since demonstrated superior efficacy compared to first-generation EGFR TKIs in the upfront setting, with improved progression-free and overall survival[2–4]. The mechanisms of acquired resistance to upfront osimertinib are still emerging, but acquired EGFR second-site mutations at EGFR C797S (the binding site of osimertinib to EGFR) are some of the commonly identified causes[5,6].

Dacomitinib is a pan-HER tyrosine kinase inhibitor that has also demonstrated efficacy as a first-line treatment for patients with *EGFR*-mutant lung cancers[7]. In the ARCHER 1050 study, dacomitinib was compared to gefitinib as the first-line treatment[8], resulting in a median progression-free survival of 14.7 months for dacomitinib compared to 9.2 months for gefitinib, leading to dacomitinib's approval in this setting. The most common mechanism of resistance to first- and second-generation EGFR inhibitors is the acquisition of a second-site mutation, EGFR T790M[9]. We, therefore, hypothesized that dacomitinib and osimertinib combination therapy might be an effective first-line treatment for patients with advanced *EGFR*-mutant lung cancers by preventing the spectrum of acquired EGFR mutations observed. Specifically, osimertinib would be effective in the presence of EGFR T790M while dacomitinib would be effective in the setting of EGFR C797S. This hypothesis is consistent with previous studies demonstrating that combination targeted therapies can delay the emergence of acquired resistance in *EGFR*-mutant lung cancers[10,11].

The identification of an optimal drug-dosing schedule of targeted therapy combinations remains a challenge due to a large number of possible drug administration schedules as well as overlapping toxicity profiles. To address this problem, several computational strategies have been proposed[12–15]. Most approaches have adopted a tumor evolution model with resistant cell clones in different patient groups[13,16,17], yet few account for the complex pharmacokinetics and the inter-patient variability of drug concentrations identified from large patient cohorts. Here we developed a predictive modeling platform that encompasses tumor heterogeneity and inter-subject variability of plasma drug concentrations to predict tumor evolution under different treatment schedules. Using this approach, we identified an osimertinib–dacomitinib combination strategy predicted to maximally decrease tumor volume while resulting in a tolerable toxicity profile, and incorporated these results into an ongoing phase 1 clinical trial (NCT03810807)[18] described herein.

## Results

### A predictive modeling framework evaluating tumor responses under different dosing strategies

To investigate the effectiveness of different combination dosing strategies, we designed a computational modeling framework to predict tumor cell number and composition during treatment with dacomitinib and osimertinib combination therapy (Fig. 1a and "Methods" section). The modeling framework considers a change in the concentrations of multiple drugs over time while taking into account inter-patient variability in pharmacokinetics as well as tumor clonal prevalence—i.e., the tumor cell composition over time. The first constituent of our modeling ensemble is the tumor clonal prevalence model, which predicts the number of cells sensitive or resistant to each individual drug over time. The model is based on a multi-type stochastic birth and death process (Fig. 1b). In this model, sensitive cells can give rise to drug-resistant cells with a specified probability per cell division for each resistance mechanism. Each cell lives for a random time period until either a mitosis or apoptosis event happens; the birth and death rates of a cell depend on the cell's mechanism of resistance as well as the concentration of each drug present at the time of cell division. The second constituent of the modeling framework, shown in Fig. 1a, is made up of population pharmacokinetic models, which allow capturing the variability of drug levels in the plasma of patients as well as the absorption and elimination kinetics over time. As the drug concentration in plasma increases, tumor cells proliferate at a slower rate, thereby making birth rates dependent on the drug concentration. The last constituent of the modeling ensemble is a pre-specified toxicity constraint, providing information on how much of each drug is tolerated in the clinic without dose-limiting side effects.

### Viability assays of EGFR-m cell lines under varying drug concentrations

In order to parameterize the computational modeling platform outlined above, we obtained proliferation rates from CellTiter-Glo (CTG) experiments using PC9 parental cell lines and drug-resistant PC9-derived cell lines harboring different mechanisms of acquired resistance (Supplementary Fig. S1 and "Methods" section). Most drug-resistant cell lines were generated by extended treatment with osimertinib or dacomitinib until resistance developed, while the PC9 C797S cell line was engineered (see "Methods" section). For the viability experiments, cells were treated with various doses of osimertinib and dacomitinib and observed for 24, 48, and 72 h, before cell plates reached confluency (Supplementary Fig. S1 and "Methods" section). Cell counts were obtained using calibration curves from the CTG experiments for each condition (Supplementary Fig. S2 and "Methods" section). Similar experiments were performed with the drug-resistant cell lines. We then obtained the growth rates of individual cell types during treatment with specific drug concentrations as the slope of a linear regression of the cell count on the log scale against time. PC9 cells, which harbor the EGFR exon 19 deletion, were found to be sensitive to both drugs, showing a significant decrease in growth rate at 10 nM osimertinib (difference in slopes of 0.0092 log-cells h$^{-1}$ between 5 and 10 nM osimertinib, $p = 0.0019$) and 0.5 nM dacomitinib (difference in slopes of 0.01685 log-cells h$^{-1}$ between 0.5 and 0.343 nM dacomitinib, $p < 0.0001$, Fig. 1c).

Comparable to previous findings[14], we observed that some cells with mutations conferring resistance had a lower fitness in the absence of treatment compared to the parental cell line (Fig. 1c and Supplementary Fig. S2). PC9-DRH cells, which harbor both the single-mutant (exon 19 del) and double-mutant (exon 19 del/T790M) alleles, were found to be resistant to dacomitinib even at 250 nM, but responded to osimertinib at 37 nM and higher concentrations ($-0.007$ log-cells h$^{-1}$ decrease in slope, $p = 0.017$), agreeing with the IC50 value of 51.2 nM (Supplementary Fig. S3a), and consistent with the correlation of viability sensitivity with suppression of downstream signaling and induction of apoptosis (Supplementary Fig. S3c). PC9R-NRAS cells, which harbor the NRAS-Q61K mutation, were resistant to both drugs since the cell count increased at a non-significantly different rate as that of the control ($p = 0.9768$, Supplementary Fig. S2b). Lastly, PC9 C797S cells were sensitive to dacomitinib starting at 27 nM (0.0119 log-cells h$^{-1}$ decrease in slope compared to 10 nM, $p = 0.020$) but were resistant even to high doses of osimertinib ($p = 0.353$, Fig. 1b), in line with an

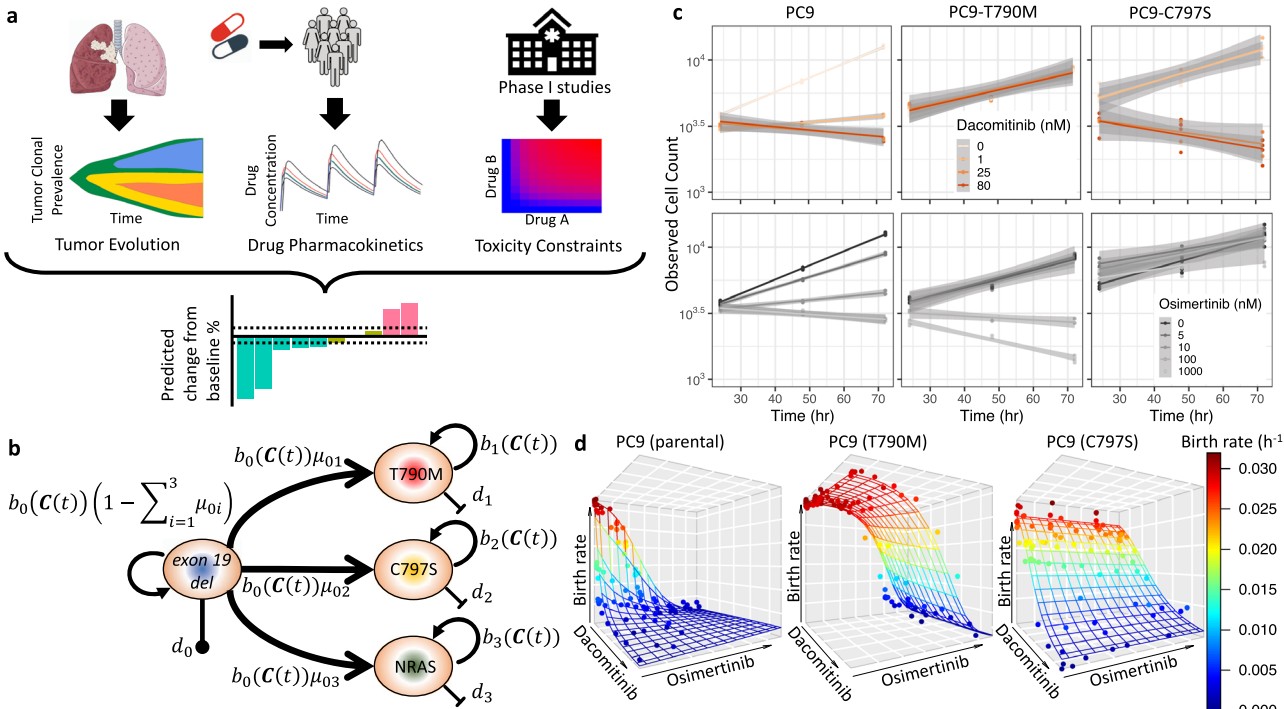

**Fig. 1 Overview of the computational modeling framework and its parameterization. a** The model ensemble consists of a tumor evolution model of multiple cell types, population drug pharmacokinetic model, and toxicity constraints and can be used to identify the most favorable therapy schedules for osimertinib and dacomitinib combination treatment. The waterfall plot represents predicted patient responses for a given dosing regimen. **b** The tumor evolution model. Each resistance mechanism arises in a one-step process. Each cell type, *i*, has its own drug-dependent birth rate and constant death rate, $b_i(\mathbf{C}(t))$ and $d_i$, respectively. The drug concentrations of dacomitinib, $C_D(t)$, and osimertinib, $C_O(t)$, were modeled as a function of time *t*. The vector of two drug concentrations, $\mathbf{C}(t) = [C_D(t), C_O(t)]$, served as the input for the multivariate birth function, $b_i$. Under a particular drug-dosing schedule, the rates are therefore time-dependent. A mutation from the sensitive cell type to type *i* occurs at rate $\mu_{0i}$ per cell division. **c** Total cell counts from CellTiter Glow (CTG) experiments during osimertinib (gray-scale lines) and dacomitinib treatment (red-scale lines). The slope of each line provides the estimated growth rate for a given cell type and drug concentration. Source data are provided as a Source Data file. **d** Birth rates of cells during combination therapy. Points represent the estimated growth rates from **c** minus death rates and the contour is the predicted birth rate as a function of dacomitinib and osimertinib concentration.

independent assessment of IC50 values of 11.7 nM and >1 mM (Supplementary Fig. S3a), respectively. This finding is also consistent with previous reports comparing sensitivity and suppression of downstream signaling in response to other second and third-generation EGFR inhibitors[19]. Based on this data, we assessed whether there was a synergistic or antagonistic interaction between osimertinib and dacomitinib with the methods of isoboles and observed no significant interaction ($p = 0.8517$ for PC9 cells, Supplementary Fig. S3b)[20].

Using this data, we found that cell types sensitive to one or more drugs had a growth rate that displayed a sigmoidal relationship with the concentration of the effective drug, whereas cells resistant to treatment exhibited a constant growth rate across drug concentrations. We then estimated the growth rates for all cell types during treatment with one or both drugs (Fig. 1c and Supplementary Table S1). These estimated growth rates of each cell type were used to build a dose-response landscape across concentrations of both drugs (Fig. 1d and Supplementary Figs. S5–S8).

Previous experiments suggested that the death rate of PC9 cells treated with erlotinib, a first-generation EGFR TKI, was almost unaffected by increasing drug doses[14,21]. Therefore, we considered that death rates of PC9-derived cells were constant and only the birth rate was dependent on drug concentration. Nevertheless, our results were robust even under the assumption of a death rate that increased with the drug concentration (Supplementary Fig. S4).

**Simulation of population pharmacokinetic models.** Population pharmacokinetic (popPK) models are complex dynamical systems that describe a drug's concentration throughout the body over time (see "Methods" section). An essential element of popPK models is inter-subject variability in pharmacokinetic parameters, which allows users to assess the variability in drug concentration between subjects. To correctly predict drug concentrations over time in our modeling ensemble, we utilized popPK models for osimertinib and dacomitinib; both models are specified as a two-compartment system with first-order absorption kinetics (Fig. 2a). The popPK model for dacomitinib was provided by Pfizer[22]. Major predictors of PK variability of dacomitinib were body weight, serum albumin level, and ethnicity, but the elimination of dacomitinib from the central and peripheral compartments was noticeably slower than that of osimertinib (Fig. 2b). The popPK model for osimertinib[23] was built using data from two NSCLC studies and one small study of healthy patients to whom osimertinib was administered orally (Fig. 2c). The daily dose of osimertinib varied from 20 to 240 mg in almost half of the patients in the three combined studies, whereas the rest of the patients received 80 mg daily. Major predictors of PK variability were bodyweight, baseline serum level, and ethnicity.

Osimertinib and one of its metabolites, AZ5104, are best represented by a two-compartment PK model[23]. AZ7550, a metabolite of osimertinib, exhibited very similar potency and profile as osimertinib against mutant and wild-type cell lines tested for inhibition of EGFR phosphorylation, whereas AZ5104

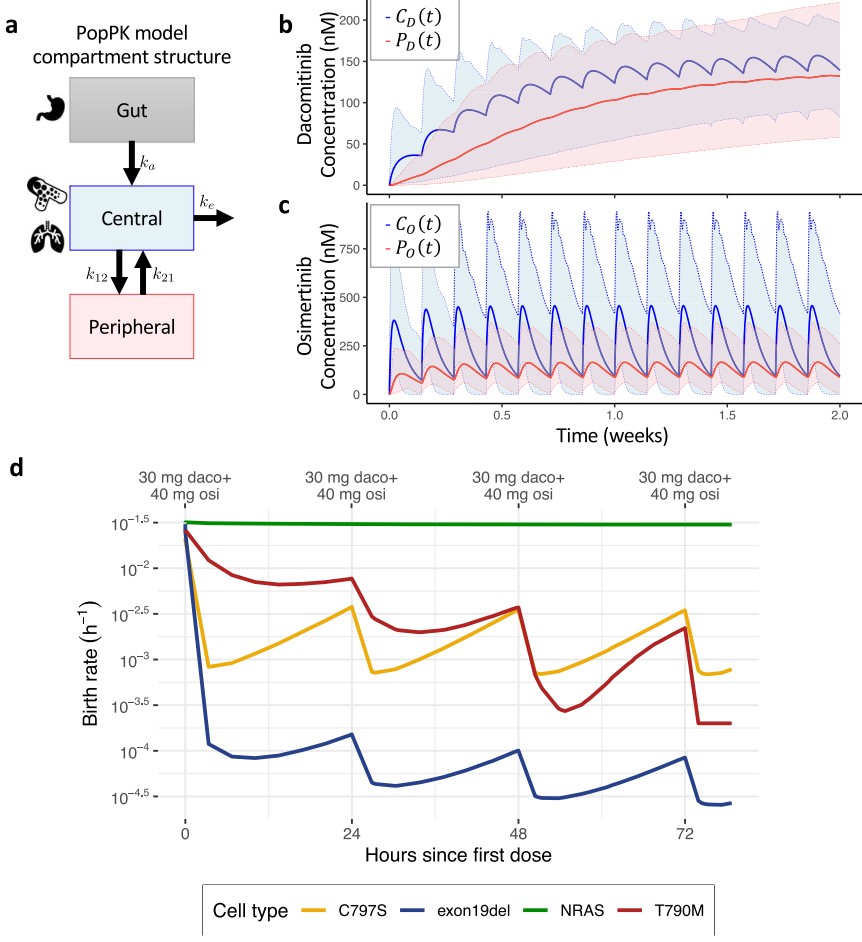

**Fig. 2 Drug pharmacokinetics with inter-subject variability and effects on tumor cell fitness. a** Schematic of a two-compartment model of drug concentration over time throughout the body. Output concentrations in nanomolars from a simulation of 30 mg QD of dacomitinib (**b**) and 40 mg QD of osimertinib (**c**) in 1000 patients. Blue and red lines correspond to drug concentration in the central ($C_D(t)$ and $C_O(t)$), and peripheral, ($P_D(t)$ and $P_D(t)$), compartment, respectively. Solid lines are median concentrations and shaded areas represent a 95% confidence interval. **d** Birth rates of four cell types during dosing with 30 mg QD dacomitinib and 40 mg QD osimertinib in one simulated individual. Doses are given every 24 h starting at hour zero.

harbored somewhat greater potency against ex19del, T790M, and wild-type EGFR cell lines[24]. However, the metabolites' concentrations are much lower than that of the parental molecule, and hence we used only the central compartment concentration of osimertinib in subsequent modeling.

Using these popPK models, we simulated drug concentrations to predict the proliferation rates of individual cell types as a function of time (Fig. 2d and "Methods" section). As a consequence, the growth rate of each cell type changes over time as the drug concentration varies in the patient due to metabolism and administration of the next dose.

**Toxicity constraints**. To avoid toxicity events, we obeyed dose limits based on the following reasoning. The MTD of dacomitinib was found to be 45 mg daily in a phase I study[25]. In later trials with planned doses of 45 mg each day (QD), the majority of patients received dose reductions or discontinued treatment as a result of adverse effects, corroborating that this is a true MTD of dacomitinib[26]. From the results of these trials, we examined only drug combinations with dacomitinib at or below 320 nM, which approximates the typical plasma concentration caused by 45 mg QD of dacomitinib (196 nM total drug exposure, the average concentration in the dosing interval, once repeated dosing has reached steady state). The recommended dose of osimertinib as a

single agent is 80 mg QD, but osimertinib has been tolerated at the 160 mg QD dose even when administered in combination[27–30]. Therefore, we allowed a maximum dose of osimertinib of 660 nM, which approximates the typical plasma concentration resulting from 160 mg QD of osimertinib (933 nM total drug exposure, the average concentration in the dosing interval once repeated dosing has reached steady state).

**In silico clinical trials identify an optimal dosing schedule**. Using the computational modeling ensemble outlined above, we then performed in silico clinical trial simulations for a large number of patients for 8 weeks, which is equivalent to two treatment cycles. In order to investigate the long-term outcomes of these trials, we also performed simulations for a year for the same simulated patients. Each simulated patient has unique PK parameters, chosen from distributions as outlined in the "Methods" section, and we performed both short-term and long-term simulations for $N = 1000$ patients to assess the impact of the PK parameters' variability on treatment performance (Fig. 3a). The trial cohorts consisted of dose escalations using a 3 + 3 design of a phase I study to identify the recommended phase 2 dose. At each dose level, we compared the outcomes in the patient simulations across different concentrations of both osimertinib and dacomitinib that are predicted to lead to comparable toxicity

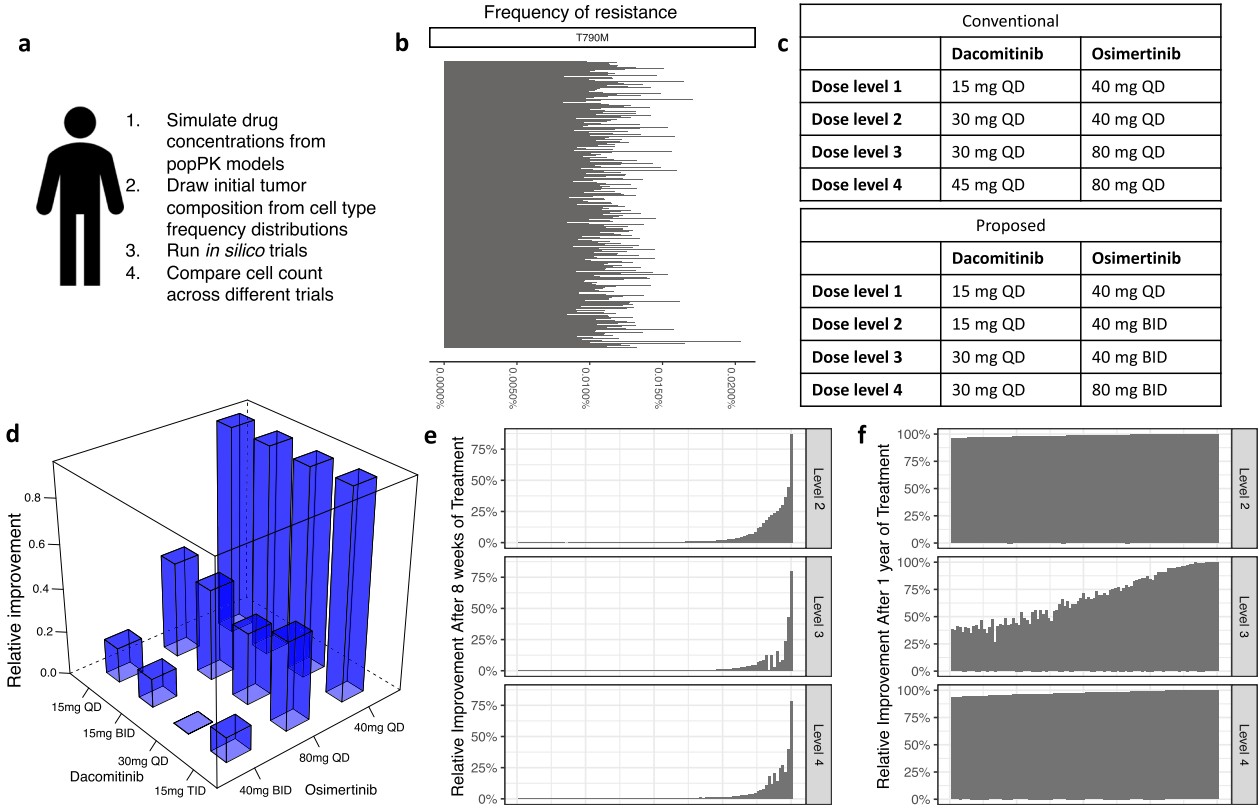

**Fig. 3 In silico clinical trials of osimertinib and dacomitinib combination therapy. a** Schematic overview of simulation steps comparison of dosing schedules for one individual. **b** Distribution of cell types before the initiation of the in silico trial over 1000 simulated patients (rows). **c** Conventional (top) and proposed (bottom) dose-escalation schedules to identify the MTD in a phase I study. **d** Comparison in outcomes of different schedules identified by their dacomitinib and osimertinib doses on the axes; the *y* axis is median improvement percentage of 30 mg QD of dacomitinib and 40 mg BID of osimertinib (proposed level 3 schedule) relative to each dose combination is shown after 1 year of treatment. **e**, **f** Waterfall plots with the relative improvement percentage of our proposed schedules compared to the conventional schedules after 8 weeks (2 treatment cycles) and 1 year of treatment, respectively, for 100 patients.

profiles. Because both dacomitinib and osimertinib are administered orally and are only available in specific quantities, we evaluated a restricted number of doses consistent with those constraints, while freely altering the timing and order of drug administration.

To accurately capture the possibility of pre-existing resistance as well as variability in the tumor volume at the time of diagnosis, for each simulated patient, we sampled clone sizes of each cell type from distributions informed by clinical information[5,14,31–36]. Thus, each patient had a unique total tumor cell number as well as the frequency of individual sensitive and resistant clones at the start of treatment (Fig. 3b and Supplementary Fig. S9, "Methods" section). Each patient was then subjected to all considered combination schedules, and the efficacy of one schedule relative to another was assessed by estimating the relative improvement. We defined the relative improvement of schedule B compared to schedule A at time *t* as:

$$\text{RI}(B, A|t) = \frac{N_A(t) - N_B(t)}{N_A(t)} \tag{1}$$

where $N_s(t)$ is the total number of tumor cells at time *t* under dosing schedule *s*. Because both drugs of interest are orally administered and doses are discrete, we investigated 15 mg (QD; twice a day, BID; or three times a day, TID), 30 mg QD, and 45 mg QD of dacomitinib in combination with 40 mg (QD or BID), and 80 mg (QD or BID) of osimertinib, the total dose depending on the dose-escalation level and previous clinical knowledge. The dose-escalation schedules that would be conventionally proposed

for a phase I/II combination study based on clinical experience would start with 15 mg QD of dacomitinib with 40 mg QD of osimertinib and if appropriate, escalate to 45 mg QD of dacomitinib with 80 mg QD of osimertinib ("conventional" in Fig. 3c).

Our modeling ensemble identified superior dose-escalation dose levels; we identified the schedule of initiating therapy with 15 mg QD of dacomitinib and 40 mg QD of osimertinib and if appropriate, escalating to 30 mg QD of dacomitinib and 80 mg BID of osimertinib (40 and 80 mg BID was superior to 80 and 160 mg QD of osimertinib, respectively) ("proposed" in Fig. 3c) to be superior to the conventional schedule. All considered schedules adhered to optimization constraints such as tolerability at each dose level and commercially available doses in tablets. Among the considered schedules, we identified the one that was predicted to minimize the number of tumor cells 1 year after the start of treatment. Additionally, we compared the schedules at two different time points: after 8 weeks of treatment (first planned follow-up of patients) and after 1 year of treatment, when resistance to the drug is typically observed. Figure 3d shows the median improvement percentage of 30 mg QD of dacomitinib and 40 mg BID (proposed level 3 schedule) of osimertinib relative to other analyzed schedules after 1 year of treatment. Specifically, we observed that 30 mg QD dacomitinib and 40 mg BID osimertinib are predicted to significantly outperform 15 mg BID dacomitinib and 80 mg BID osimertinib ($P = 0.034$), and also to outperform 15 mg TID dacomitinib and 80 mg QD osimertinib ($P = 0.03$). In contrast to 30 mg QD dacomitinib and 80 mg QD

osimertinib, we observed a marginally significant improvement after 1 year of treatment ($P = 0.051$) after multiple comparison corrections. When comparing the optimized versus conventional schedule, we observed a notable improvement in roughly 20% of the cases for level 2 through level 4 schedules after two treatment cycles (Fig. 3e), mainly driven by the suppression of T790M clones (Supplementary Fig. S10a), and an improvement in 100% of patients after 1 year of therapy (Fig. 3f). Thus, we predicted that smaller but more frequent doses of osimertinib with larger and less frequent doses of dacomitinib would result in improved outcomes in simulated subjects. We repeated our analyses for patients with a bodyweight of 90 kg (median is 70 kg) and observed that our proposed schedules would still fare better after 1 year of treatment, but a quarter of patients would benefit from taking 80 mg osimertinib QD over 40 mg osimertinib BID in level 3 of the dose-escalation (Supplementary Fig. S10b). Nevertheless, the improvements of these patients were relatively low (median 5%) compared to those of patients who benefited from 40 mg osimertinib BID (median about 50%). Thus our proposed dose-escalation schedule outperformed the conventional schedule across all simulated patients, even after incorporating inter-patient variability in drug concentration and tumor heterogeneity. This result is encouraging for clinical implementation because customization of schedules for each individual patient was able to be avoided.

**Pharmacokinetic variability contributes to the heterogeneity in treatment response.** The popPK models of both dacomitinib and osimertinib contain parameters such as the volume of distribution and clearance rate distributions for the central and peripheral compartments, absorption rates, and transfer/elimination rates. In our in silico clinical trial studies, these PK parameters were considered to be constant across different doses but unique to each patient. The PK parameters are log-transformed linear mixed effect models composed of clinical variables, such as body surface area or albumin levels in the blood (see "Methods" section). We then sought to analyze the association between the different PK parameters and the estimated relative improvement percentages based on simulation results of the in silico trials. To this end, we fixed the clinical predictors at the median and used a Spearman correlation estimator for comparisons. We found that there was a statistically significant negative correlation ($p < 0.0001$) between the volume of distribution of the central compartment of osimertinib and the estimated improvement of the proposed versus conventional schedule. This observation suggests that patients with low estimates for this PK parameter may have a better outcome when treated with the proposed schedule as compared to other patients. Furthermore, the absorption rate and clearance of distribution of the central compartment of osimertinib were significantly positively associated ($p < 0.0001$) with the predicted relative improvement of the proposed schedule. In general, we found that PK parameters of osimertinib were more predictive of an improvement than those of dacomitinib. To quantify this finding, we performed random forest analysis to rank the importance of parameters on the estimated improvement of our proposed schedule compared to the conventional schedule (Fig. 3d). The parameters used in the random forest were the pre-existence of resistant mechanisms before treatment, initial tumor size, and all pharmacokinetic parameters. This analysis identified that after 2 weeks of therapy, the prevalence of the NRAS mutation before treatment, $p_{NRAS}$, was the most important parameter (Supplementary Fig. S11a). However, the importance of this parameter decreases after 1 year of treatment (Supplementary Fig. S11b), suggesting that over time, the pharmacokinetic features of a patient or the pre-existence of T790M

become more important for determining response to combination therapy than the pre-existence of NRAS mutations. We further explored the relationship of PK parameters with resistant subclones and observed that the clearance of the central and peripheral compartments in the osimertinib popPK model were positively correlated with the frequency of T790M+ subclones during treatment (Supplementary Fig. S11c, $p < 0.001$, Spearman test with Bonferroni correction). With respect to the frequency of C797S+ subclones, the clearance of both compartments in the dacomitinib model was positively associated with subclone count, whereas the absorption rate of dacomitinib was negatively correlated with C797S cell counts (Supplementary Fig. S11d).

**Model validation using long-term in vitro cell culture assays.** To validate our predictive modeling platform, we performed duration of response (DoR) in vitro cell culture assays for several specific dosing schedules (see "Methods" section). Because these assays are not performed on organisms in which drug concentrations change substantially over time, we used fixed drug concentrations for the validation experiments; thus we utilized the conventional and proposed dosing schedules, shown in Fig. 3c, to derive the drug concentrations applied in the validation experiments by estimating the median drug concentration at steady state (Supplementary Fig. S12a). The derived in vitro equivalents of 45 mg of dacomitinib and 80 mg of osimertinib are 4.40 and 10.38 nM, respectively. Additionally, 2.94 nM of dacomitinib and 5.19 nM of osimertinib are analogous to 30 mg QD of dacomitinib and 40 mg QD of osimertinib (conventional dose level 2), whereas 1.47 nM of dacomitinib and 10.38 nM of osimertinib correspond to the proposed dose level 2 (Fig. 3c). The conventional dose level 4 and proposed dose level 4 yielded 4.40 nM dacomitinib + 10.38 nM osimertinib and 2.94 nM dacomitinib + 20.75 nM osimertinib, respectively. Lastly, 40 mg BID osimertinib produces the same in vitro concentration as 80 mg QD osimertinib, so we were unable to compare dose level 2 schedules.

We used several starting conditions for these experiments. The RPC9-CL9 clone is composed of 90% allele frequency exon 19 deletion and 10% allele frequency exon 19 deletion as well as T790M. Mixed pools composed of PC9 cells and RPC9-CL9 cells (at ratios 10–1 and 100–1) were also investigated in these studies.

When comparing the experimental results with the modeling predictions based on parameter estimates obtained in Fig. 1, we found that the predicted tumor cell counts accurately described the validation data in terms of identifying the superior regimen at each level of the dose-escalation process. Generally, for long-term assays (i.e., more than 50 days), the predicted cell counts were higher than those observed, perhaps suggesting a lower true mutation rate or some other intrinsic decrease in cell growth due to their environment (Supplementary Fig. S12b), which has been observed in previous studies under similar conditions (see Supplementary Fig. S8E in ref. [30]). In the PC9 to RPC9-CL6 ratio pool of 10:1, we observed an agreement in cell number under different drug concentrations at three different time points (Fig. 4a). On day 30 of the assays, we correctly predicted the superiority of 2.94 nM of dacomitinib + 20.75 nM of osimertinib (corresponding to 30 mg QD of dacomitinib and 80 mg BID of osimertinib) over 4.40 nM of dacomitinib + 10.30 nM of osimertinib (corresponding to 45 mg QD of dacomitinib and 80 mg QD of osimertinib), which was also observed in the validation experiments ($p = 0.0159$). In the PC9 to RPC9-CL6 ratio pool of 100:1, our model also proved to be accurate in terms of cell numbers on days 10, 20, and 30 of treatment (Fig. 4b). On day 10, all schedules seemed to have a similar effect on cell numbers, while the disparity between the schedules' efficiency became clearer on day 30, where our model correctly ranked the first 3 schedules. However, one of the

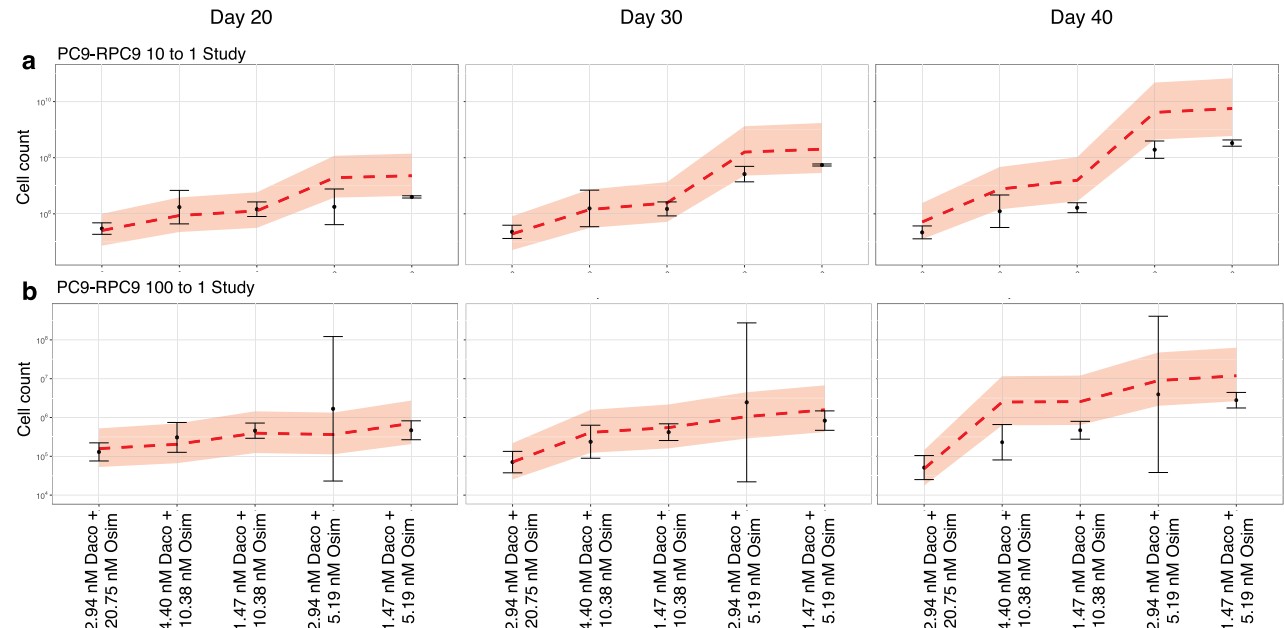

**Fig. 4 Longitudinal validation experiments in mixed cell pools at days 20, 30, and 40 of treatment.** Predictions and interquartile ranges from mathematical modeling predictions are shown in a dashed red line and shaded regions, respectively. Observations with two standard errors are shown in black dots and error bars, respectively. The RPC9 (RPC9-CL6) clone is composed of 90% exon 19 del cells 10% exon 19 del and T790M alleles. **a** PC9-RPC9 cell pool in 10–1 ratio. Our predictions ranked schedules correctly, detecting a difference between the two best schedules at day 30 ($p = 0.0159$, two-sided $t$-test with $n = 6$ biological samples). **b** PC9-RPC9 cell pool in 100–1 ratio. Our predictions ranked schedules correctly except for the worst 2 schedules, but the difference between the schedules was not statistically significant in the observed data ($p = 0.294$, two-sided $t$-test with $n = 6$ biological samples). Source data are provided as a Source Data file.

| Table 1 Dose levels in the dose-escalation study. | | | | |
|---|---|---|---|---|
| **Dose level** | **Osimertinib** | **Dacomitinib** | **# of Patients enrolled** | **DLT?** |
| 1 | 40 mg daily | 15 mg daily | 3 | None |
| 2 | 40 mg twice daily | 15 mg daily | 3 | None |
| 3 | 40 mg twice daily | 30 mg daily | 6 | None |
| 4 | 80 mg twice daily | 30 mg daily | N/A | N/A |
| RP2D dose expansion | 40 mg twice daily | 30 mg daily | 10 | None |

schedules (2.94 nM dacomitinib + 5.19 nM osimertinib) displayed an abnormally high variability, which resulted in a non-significant difference from a schedule expected to fare worse. Furthermore, we found that EGFR pathway suppression and induction of apoptosis in PC9 and RPC9-CL6 cells were consistent with the long-term viability effect of the combination (Supplementary Fig. S3c). PC9 cells with the NRAS mutation, whose estimated birth rate is almost constant at any dosing of dacomitinib and osimertinib (Fig. 2d), displayed no change in growth as expected (Supplementary Fig. S12d). Thus, our model was successfully validated in vitro.

**A phase I study using the modeling-derived dosing schedule of osimertinib and dacomitinib.** Based on our modeling predictions, we incorporated the predicted dosing schedule in a phase 1 study at Memorial Sloan Kettering Cancer Center (see "Methods" section). We enrolled 22 patients: 12 patients in initial dose escalation and 10 patients in an expansion cohort at the recommended phase 2 dose. In dose escalation, patients received doses ranging from dacomitinib 15–30 mg daily and osimertinib 40 mg daily to twice daily (Table 1). The baseline characteristics of the study population are shown in Table 2. Patient characteristics

| Table 2 Baseline patient characteristics. | |
|---|---|
| Characteristic | $n = 22$ |
| Median age, years (range) | 65 (36–78) |
| Sex | |
| Female | 15 (68%) |
| Male | 7 (32%) |
| KPS (%) | |
| ≥90 | 16 (73%) |
| 80 | 6 (27%) |
| Smoking status | |
| Former (pack years range) | 12 (2–30) |
| Never | 10 (45%) |
| *EGFR* sensitizing mutation | |
| L858R | 8 (36%) |
| Exon 19 deletion | 13 (59%) |
| L861Q | 1 (5%) |
| Brain metastases | |
| No | 13 (59%) |
| Yes (untreated) | 9 (9) |

**Table 3 Grade 1–3 treatment-related adverse events.**

| Adverse event | Treatment-related toxicities | | | | |
|---|---|---|---|---|---|
| | Grade 1 | Grade 2 | Grade 3 | Total | Incidence |
| Diarrhea | 13 | 4 | 2 | 19 | 86% |
| Rash acneiform | 8 | 7 | | 15 | 68% |
| Mucositis oral | 5 | 7 | 1 | 13 | 59% |
| Dry skin | 9 | 2 | 1 | 12 | 55% |
| Anorexia | 2 | 6 | 1 | 9 | 41% |
| Dysgeusia | 6 | 2 | | 8 | 36% |
| Fatigue | 5 | 3 | | 8 | 36% |
| Paronychia | 6 | 2 | | 8 | 36% |
| Rash maculopapular | 3 | 4 | | 7 | 32% |
| Pruritus | 5 | 1 | | 6 | 27% |
| Weight loss | 4 | 1 | 1 | 6 | 27% |
| Alopecia | 4 | | | 4 | 18% |
| Nausea | 2 | 2 | | 4 | 18% |
| Skin infection | 2 | 2 | | 4 | 18% |
| Cough | 1 | 2 | | 3 | 14% |
| Dry eye | 2 | 1 | | 3 | 14% |
| Rhinorrhea | 3 | | | 3 | 14% |

were in line with typical patients with newly diagnosed *EGFR*-mutant lung cancers (predominantly never/light smokers, women). Forty-one percent had brain metastases at diagnosis, and all nine were untreated with radiation or surgery prior to study start. The patients were enrolled from January 2019 to May 2020 and the database lock was April 12th, 2021.

All patients were evaluable for toxicity assessment. There were no grade 4 or 5 toxicities and the most common treatment-related adverse events were grade 1–2 diarrhea and rash (Table 3). 15 patients had a dose reduction of dacomitinib, of which 5 patients also had dose reductions of osimertinib (Supplementary Table S2). The more common toxicities resulting in dose reduction were rash ($n = 5$), mucositis ($n = 3$), and diarrhea ($n = 3$). Three patients discontinued study treatment with intolerable grade 1–2 toxicity (Supplementary Table S2). The decision was made not to dose escalate beyond dacomitinib 30 mg daily and osimertinib 40 mg twice daily (dose level 3) due to feasibility. Although there were no dose-limiting toxicities (DLT), the frequency of grade 1–2 drug-related toxicity was high and intolerable for patients over the long treatment duration expected. The recommended phase 2 dose was dacomitinib 30 mg daily and osimertinib 40 mg twice daily. Patients in the dose-expansion arm received dacomitinib 30 mg daily and either osimertinib 40 mg twice daily or osimertinib 80 mg once daily depending on insurance coverage.

The study is complete and has met its primary endpoint of identification of the recommended phase 2 dose of the combination. Patients can continue to remain on study if they are clinically benefitting from treatment and to further assess long-term toxicity. Efficacy endpoints remain immature. The median follow-up is 15.1 months. Of the 22 patients treated, 21 have had radiographic assessments of their disease. 16 of 22 have had a confirmed partial or complete response to treatment resulting in an overall response rate of 73%. Six patients have come off treatment for disease progression and 1 patient withdrew from the study unrelated to toxicity or progression. Follow-up is ongoing.

## Discussion

In this paper, we established a predictive modeling platform that, when parameterized using information from in vitro cell line assays, can be used to identify optimal combination dosing schedules for osimertinib and dacomitinib treatment for patients with *EGFR*-mutant NSCLC. Our approach suggests that higher but less frequent doses of dacomitinib and lower but more frequent doses of osimertinib would yield superior results. This observation is due to the pharmacodynamic and pharmacokinetic profiles of the individual cell types and drugs under consideration. Dacomitinib has a narrower therapeutic index than osimertinib, and therefore it is necessary to maintain a sufficiently high concentration of dacomitinib in plasma, which is reached by administering a larger dose. Furthermore, the drug's pharmacokinetics suggest a slow depletion in plasma, making frequent doses less necessary to maintain the drug concentration within the therapeutic window (Fig. 2b). On the other hand, osimertinib has a large therapeutic index and its concentration in plasma is depleted relatively quickly (Fig. 2c). As a result, it is not necessary to administer large doses of osimertinib since a lower dose still reaches the therapeutic window, but to maintain the drug concentration within this window, frequent dosing is required to keep pace with the drug's rapid elimination from the system. Our mathematical modeling approach yields the additional advantage of a quantitative comparison of the performance of various schedules. Our modeling approach, based on extensive human PK data and in vitro dose-response information, provided sufficient support to approve the recommended dosing regimen for a clinical trial, without the necessity of any additional in vivo experiments. We, therefore, performed a phase 1b study incorporating the dosing schedule derived from our predictive modeling platform and demonstrated the safety and tolerability of the combination, a proof of concept that could justify broader use of this technique in future phase 1 combination trials.

The rate of serious adverse events remained low and did not exceed historical experience with single-agent dacomitinib or osimertinib. Predictably, we did identify frequent low-grade rash and diarrhea, which are the most common toxicities with single-agent dacomitinib and osimertinib; these lower grade toxicities were somewhat additive with the combination. However, grade 1–2 issues resulted in dose reduction in 15 cases and trial discontinuation in 3 cases. Lower-grade gastrointestinal and cutaneous toxicities significantly impact the quality of life for patients and often lead to dose reductions especially for treatments that are expected to be efficacious over a long time period. Therefore, we ultimately established dacomitinib 30 mg daily and osimertinib 40 mg twice daily as the recommended dose for future studies. Our clinical data demonstrate that even when utilizing mathematical modeling and careful consideration of dose, combination EGFR inhibition is challenging due to consistent, lower grade gastrointestinal and cutaneous toxicities over the long treatment duration. Our mathematical model can be applied to other targeted therapies using proper data to parametrize tumor drug response. For instance, other second-generation EGFR inhibitors elicit similar responses to dacomitinib, and we, therefore, expect that our modeling platform, together with appropriate dose-response and PK data, can be applied for designing combination trials for other agents as well.

There are a few limitations to our paper. First, we lacked toxicity data from patients receiving osimertinib in combination with dacomitinib in the clinic so this could not be accounted for in our model (NCT03810807)[18]. Furthermore, our mathematical modeling does not take into account a tumor's carrying capacity or spatial constraints; we opted to use a more parsimonious model since an approach containing a carrying capacity would contain a large number of variables that may vary considerably among patients as well as environmental conditions[37].

Our modeling platform can be used to personalize treatment for individual patients. We envision that, with the advent of methodology to obtain real-time patient-specific PK information, our models can be used to identify the best dosing schedule for

each patient. Drugs that are administered intravenously or drugs that can be obtained for a variety of dose options per tablet may be better suited for individual optimization. We also aim to enhance our model ensemble by expanding toxicity modeling and combination with popPK models to predict which patients are most likely to have adverse effects from the drug combination. Another future application of interest is to identify an optimal schedule for patients with brain metastases. Osimertinib can successfully penetrate the blood-brain barrier: patients who had received 160 mg osimertinib daily had a concentration of about 7.51 nM in the central nervous system[28]. Even though the effect of dacomitinib on brain metastases has not been studied in human patients, dacomitinib has displayed brain penetration in preclinical models and thus can be active as well in clinical settings[38,39]. Finally, we intend to increase the practicality of our model by incorporating other resistance mechanisms commonly observed in treated EGFR-m NSCLC and feature other targeted therapies in combination with EGFR inhibitors.

## Methods

**Clinical trial**. The study was a phase 1 dose-escalation study of combination dacomitinib and osimertinib for patients with metastatic EGFR-mutant lung cancer. The study was approved by the Memorial Sloan Kettering Institutional Review Board; the study design and conduct complied with all required regulations and was conducted in accordance with the criteria set by the Declaration of Helsinki. There were three potential dose levels of the combination and once the MTD was identified, 10 additional patients were to be enrolled at the MTD to complete an expansion cohort. The primary objective of the study was to determine the maximum tolerated dose of the combination of dacomitinib and osimertinib. Secondary objectives included measuring best overall response, progression-free survival, and overall survival. After the maximum tolerated dose/recommended phase 2 dose was established, an expansion cohort at that dose was enrolled with the primary objective to further establish the toxicity profile of the combination. The secondary objective of the expansion cohort was to obtain preliminary efficacy data by measuring the objective response rate. Up to 34 patients could be enrolled. Patients were required to have biopsy-proven metastatic EGFR-mutant lung cancer and not to have had prior treatment with an EGFR TKI to enroll. See the study protocol for additional eligibility criteria. The first patient enrolled in January 2019 and the last patient in May 2020. The database lock was April 2021.

**Compounds, cell lines, and culture conditions**. Dacomitinib, PF-06747775, and osimertinib were synthesized via Pfizer Worldwide Research and Development (La Jolla, CA, USA). Compounds were dissolved in dimethyl sulfoxide (DMSO) (10 mM) and diluted in a cell culture medium for evaluation of cellular potency. PC9 cells were purchased from RIKEN Cell Bank (Tsukuba, Ibaraki Prefecture, Japan) and were cultured in Gibco® RPMI (Life Technologies™, Carlsbad, CA, USA) medium with 10% heat-inactivated FBS (Sigma®, St. Louis, MO, USA). PC9-DRH, harboring both the single-mutant (Del) and double-mutant (Del/T790M) alleles, is a pool of cells derived from the PC9 parental line that was selected after treatment with gradually increasing concentrations of dacomitinib up to 2 μM. PC9-DRH EGFR alleles consist of 70% Del/T790M and 30% Del. PC9-DRH cells were cultured in Gibco® RPMI medium with 10% FBS, and maintained in dacomitinib (2 μM)[40]. RPC9-CL6, harboring both the single-mutant (Del) and double-mutant (Del/T790M) alleles, is a clone of cells derived from the PC9 parental line that was selected after treatment with gradually increasing concentrations of dacomitinib up to 2 nM. RPC9-CL6 EGFR alleles consist of 10% Del/T790M and 90% Del. RPC9-CL6 cells were cultured in Gibco® RPMI medium with 10% FBS, and maintained in erlotinib (2 μM). PC9R-NRAS, harboring 40% allele frequency NRAS-Q61K mutation and sensitive to treatment with the combination of a third-generation EGFR TKI (e.g., osimertinib) plus a MEK inhibitor (e.g., selumetinib), is a pool of cells derived from the PC9 parental line that was selected after treatment with gradually increasing concentrations of a third-generation EGFR TKI PF-06747775 up to 1 uM. PC9R-NRAS cells were cultured in Gibco® RPMI medium with 10% FBS and maintained in PF-06747775 (1 μM). PC9 C797S cells were generated by introducing an EGFR Exon 19 del/C797S construct into PC9 cells via lentiviral transduction as previously described[19] and cultured in RPMI with 10% FBS.

**Cell viability assays**. Cells (2000 cells/well for PC9 or 3000 cells/well for PC9-DRH, PC9R RRAS, and PC9 C797S) were seeded in a 96-well microtiter plate and allowed to adhere overnight. The compound was added and incubated for 24, 48, or 72 h. After compound incubation, cell viability was measured utilizing CellTiter-Glo® (Promega Corporation, Madison, WI, USA) reagent following the manufacturer's instructions. The resulting luminescence signal was read using an EnVision® Multilabel Reader (PerkinElmer®, Waltham, MA, USA) plate reader.

Baseline CTG read was performed in a separate plate after overnight seeding (day 0) under the same cell seeding condition as the drug treatment plates. The concentration–response curve was plotted as log drug concentration ($X$ axis) versus percent control ($Y$ axis) using GraphPad PRISM. For the estimation of cell numbers, the ratio of CTG read per cell is calculated based on the following formula

$$\text{Ratio of CTG}_{\text{cell}} = \frac{\text{CTG}_{\text{Day 0 with cells}} - \text{CTG}_{\text{Day 0 without cells}}}{N_{\text{cells}}} \quad (2)$$

where $N_{\text{cells}}$ is the number of seeded cells.

**Immunoblotting analysis**. Cells were treated with compound for 24 h, harvested, and stored at −80 °C until further analysis. Cell pellets were lysed in lysis buffer (150 mM NaCl, 1.5 mM MgCl₂, 50 mM 4-(2-hydroxyethyl)-1-piper-azineethanesulfonic acid (HEPES), 10% glycerol, 1 mM ethylene glycol tetraacetic acid (EGTA), 1% Triton ® X-100, 0.5% NP-40) supplemented with 1 mM Na₃VO₄, 1 mM phenylmethylsulfonyl fluoride (PMSF), 1 mM NaF, 1 mM β-glycerophosphate, cOmplete Mini EDTA-free Protease Inhibitor Cocktail Tablets, and PhosSTOP prior to use. Protein concentration was determined using the BCA Protein Assay (Pierce/ThermoFisher Scientific, Rockford, IL, USA) as per the manufacturer's instructions. Ten μg of total protein was resolved by sodium dodecyl sulfate-polyacrylamide gel electrophoresis (SDS-PAGE) and transferred onto the nitrocellulose membrane. Blots were probed with primary antibodies overnight at 4 °C in the manufacturer's recommended buffer to detect proteins of interest. After incubation with secondary antibodies, signals were visualized by chemiluminescence (Pierce/ThermoFisher Scientific) on a FluorChem™ Q digital imager (Protein Simple™, Santa Clara, CA, USA). Antibodies against EGFR (4267), p-EGFR Y1068 (3777), extracellular signal-regulated kinase (ERK) (9102), phosphorylated-ERK (p-ERK) T202/Y204 (9101), Akt (4691), pAktS473 (4060), S6 ribosomal protein (S6) (2217), phosphor-S6 ribosomal protein S235/236 (pS6) (4858), cleaved PARP (9541) and glyceraldehyde-3-phosphate dehydrogenase (GAPDH) (2118) were purchased from Cell Signaling Technology® (Danvers, MA, USA). Dilutions used for experiments are shown in Supplementary Table S3.

**Mathematical/statistical modeling**. We used R version 3.5 for all analyses[41]. For each cell type and drug combination, we used OLS regression with a logarithmic transformation and regressed cell count against time to obtain a growth rate (Fig. 1b, c). After obtaining these growth rates, we used linear regression with transformed outcomes to model growth rates given the concentrations of dacomitinib and osimertinib. The transformation and the regression are given by

$$\log\left(\frac{y_i - y_{\min}}{y_{\max} - y_i}\right) = C_i \boldsymbol{\beta} + \epsilon_i, \epsilon_i \sim N(0, \sigma^2) \quad (3)$$

where $y_i$ is the growth rate at observation $i$, and $y_{\max}$ and $y_{\min}$ are respectively the maximum and minimum growth rates. $C_i$ is a vector of transformed drug concentrations (selected when finding the best model fit), while $\boldsymbol{\beta}$ is a vector containing the fixed effects with respect to $C_i$ (Supplementary Figs. S5C, S6C, S7C, and S8D). We obtained the birth rate by adding the death rate to the predicted growth rate and assumed that death rates were time-invariant.

To model tumor evolution, we used a multi-type branching process with drug-dependent, and hence time-dependent, birth and death rates[12,42]. We assumed that resistance mechanisms are not mutually exclusive within a tumor (i.e., each patient may exhibit more than 1 resistance mechanism), but they are mutually exclusive within a cell due to lack of experimental data based on cell lines harboring multiple mutations.

We assumed that the resistance mutations T790M and C797S were not mutually exclusive, but we are aware that the allelic context is important[19]. If the C797S and T790M mutations are in *trans*, cells are resistant to third-generation EGFR TKIs, but are sensitive to a combination of first and third-generation TKIs. If the mutations are in *cis*, no single EGFR TKI alone or in combination can suppress activity. If C797S develops in cells without T790M (observed when third-generation TKIs are administered as first-line therapy), the cells are resistant to third-generation TKIs, but retain sensitivity to first-generation TKIs. Our tumor evolution assumptions model the last type of resistance mechanism (C797S without T790M).

**Simulation of population pharmacokinetic models**. We used popPK models to simulate drug concentrations in various individuals[43]. PopPK models, which tend to be multi-compartmental, focus on the sources of variability in drug concentrations among individuals who are the target patient population receiving clinically relevant doses of a drug of interest[44,45]. These methods possess key advantages for characterizing PK across multiple studies, for exploring PK variability due to intrinsic factors (e.g., age, sex, race, mutation status) and extrinsic influences, and for informing dosage adjustments based upon these influences[46]. We opted to use popPK models to incorporate between-subject variability commonly observed in drug studies. We assumed that the drug concentration in the tumor is comparable to the drug concentration in the central compartment, which is typically represented as the drug concentration in plasma or highly vascular tissue[47].

To simulate the plasma concentration of dacomitinib, we used a popPK model constructed by Pfizer using a two-compartment structure[22]. We used a published popPK model for osimertinib that is based on a two-compartment linear structural model[23]. In the latter approach, a first-order conditional estimation (FOCE) with interaction was used to derive the mean and variance estimates. The FOCE method was then used to approximate the Hessian matrix due to the difficulty of direct computation of second-order derivatives[48].

We simulated drug concentrations simultaneously using the multi-platform package Ubiquity[49]. Because PK parameters are functions of clinical variables, we set these variables to the reported median from a previous study[23]. In subsequent simulations, we also simulated subjects with different predictors and observed effects on tumor growth.

Let $\boldsymbol{\eta} = \left( \eta_{CL,daco}, \eta_{V,daco}, \eta_{CL,osi}, \eta_{V,osi} \right)^{T}$ be a vector composed of the distributions of the random effects used in the PK models of dacomitinib and osimertinib. By the models' descriptions, we have that $\boldsymbol{\eta} \sim \mathrm{MVN}(\mathbf{0}, \boldsymbol{\Omega})$. The studies of the popPK models we used in the simulations reported $\mathrm{Cov}\left( \eta_{CL,daco}, \eta_{V,daco} \right)$ and $\mathrm{Cov}(\eta_{CL,osi}, \eta_{V,osi})$. Because we had no estimates for the covariances between the drugs' random effects, we identified correlations that would allow the covariance matrix to remain positive semi-definite. We simulated the following scenarios:

$$\rho_{CL} = \{0, 0.15, 0.3\} \text{ and } \rho_{V} = \{0, 0.2, 0.5, 0.8\} \tag{4}$$

where $\rho_{CL} = \mathrm{Cor}(\eta_{CL,daco}, \eta_{CL,osi})$ and $\rho_{CL} = \mathrm{Cor}(\eta_{V,daco}, \eta_{V,osi})$. Only the listed correlations yielded a positive semi-definite covariance matrix. There were larger improvements with our proposed schedules compared to the conventional schedules assuming high correlations between the random effects of the drugs, but there was still an improvement when we assumed independence between the popPK parameters.

**Initial populations and mutation rates**. A prior study reported that the median baseline sum longest diameter (BSLD) of lung cancers is 7.5 cm, with an IQR 4.5–11.7 cm and the lowest observed (and detectable) value being 1 cm of diameter at diagnosis[50]. To convert the tumor size at diagnosis to cell numbers, we used the equivalency of 1 cm$^3$ = 10$^8$ tumor cells as suggested by prior literature for most tumors[51,52]. We used this information of tumor diameters to create the following truncated distribution:

$$\mathrm{BSLD} = X\mathbf{1}(X \geq 1) \tag{5}$$

where $X \sim N(7.5, 39) \, \mathrm{cm}^3$, $\mathbf{1}()$ is the indicator function, and sampled zeroes were discarded. We set the truncation at 1 because the minimum detectable diameter of a lung tumor[50] is 1 cm. We then converted those sampled diameters to volume ($V$) in cm$^3$ and multiplied by 10$^8$ to obtain the number of cells:

$$V = \frac{4\pi \left( \frac{\mathrm{BSLD}}{2} \right)^3}{3} \tag{6}$$

To sample the initial tumor cell count of each resistant cell type for each simulated patient, we used estimates based on prior literature; T790M in newly diagnosed patients[14,31–33] has been suggested to occur with a frequency of 1 out of 3 million cells[14]. Using droplet digital PCR (ddPCR), it was shown that ~80% of patients with NSCLC harboring EGFR-activating mutations harbored the T790M mutation at an ultra-low allele frequency between 0.001% and 0.1% pre-treatment based on an analysis of 373 patients[32]. Therefore, about 80% of simulated patients had pre-existing T790M clones with a mutation rate (from drug-sensitive cell to T790M cells) sampled from a normal distribution centered at $10^{-7}$ (Supplementary Fig. S9a). NRAS mutations are less common than other resistance mechanisms and arise in about 1% of NSCLC patients[34]. There is not enough quantitative data of NRAS pre-existence in untreated NSCLC; therefore, we assumed that the presence of NRAS-positive clones in the simulated pre-treatment tumors is rarer than that of T790M by two orders of magnitude and present in only 1–5% of simulated patients. The NRAS mutation rate is considered to be substantially smaller than that of T790M, and thus we sampled mutation rates from a distribution centered at $10^{-9.5}$ but with a smaller variability to avoid extremely small mutation rates. C797S arises in about 10% of NSCLC patients in the US, but in 40% of NSCLC patients who have developed T790M-driven resistance when treated with first-generation EGFR TKIs[5]. However, the prevalence of C797S mutations among patients with third-generation EGFR TKIs as first-line therapy has remained low; thus, we assumed no pre-existence and a mutation rate distribution of one order of magnitude lower than that of T790M (Supplementary Fig. S9b). Lastly, we did not restrict pre-existing mutations to be mutually exclusive within a tumor; in other words, a simulated pre-treated tumor could either have T790M-positive cells and NRAS-positive cells, only one of these mutations present, or neither.

**Trial simulations and comparison of schedules**. We used a multi-type birth and death branching process with continuous birth and death rates to simulate the evolutionary dynamics of a tumor[12]. Each drug-sensitive cell lives for a random amount of time that follows an exponential distribution before mitosis or death.

During mitosis, a cell has a small probability of developing a resistance mechanism to one or more drugs. After a mutation has arisen, the newly emerged drug-resistant cell may have different birth and death rates than those of the parent cell. In our simulations, the birth rate is dependent on drug concentration which is dependent on time; the death rates are assumed to be constant. Stochastic processes are computationally expensive when the initial cell count is large. As a result, we calculated the expected number of cancer cells in different clones under treatment using a previously developed approach for situations when the stochastic approach was not feasible[12].

We considered schedules that adhere to two optimization constraints: (1) an approximation of the maximum amount of tolerated drug and duration of exposure using clinical experience, and (2) schedule options depending on commercial dose availability. Osimertinib is available in 40 mg and 80 mg tablets, whereas dacomitinib is available in 15, 30, and 45 mg tablets. Thus, we used these doses in our simulations and varied the frequency of administration while complying with the first constraint.

After estimating cell counts under a specific dosing schedule at time $t$, we compared different schedules by estimating the relative improvement defined as in Eq. 1. We reported the comparisons of the total cell count to identify the schedule that performed better across all subclones.

**Duration of response study**. Cells (PC9, RPC9-CL6, PC9R-NRAS, and mixed PC9/RPC9-CL6) were seeded in T25 flasks (three flasks per condition). When cells reached a density of estimated cell confluence as ~40%, dacomitinib and/or osimertinib at the indicated concentration were added to the cells and the time is recorded as day 0. Cells were fed fresh medium and drug every week and cell confluence was monitored on a daily basis. When cell growth reached ~75% confluence, cells were harvested, counted, and 1/6 of the collected cells were used to seed a fresh T25 flask. Once growth reached ~75% confluence again, this process was repeated until cells reach 32 times the initial cell seeding (5 doubling time). The predicated viable cell number ($N_P$) was calculated using:

$$N_P = N_{P_n} \times 6^{n-1} \tag{7}$$

where $N_{P_n}$ is the live total cell count at passage $n$.

**Determining concentrations for in vitro validation**. Using drug concentrations that correspond to the schedules we analyzed, we estimated the average of the median plasma concentration at the steady-state defined as

$$\bar{C} = \frac{\int_0^\tau C(t_{ss})_{median} dt_{ss}}{\tau} \tag{8}$$

where $C(t)_{median}^{ss}$ is the median concentration from our simulations at steady state (reached after 15 days from initial dose). We adjusted for plasma protein binding by multiplying average plasma concentration with the unbound fraction in human serum for each drug, which provided the following concentrations

$$\mathrm{UC}_{daco} = \overline{C_{daco}} \times 0.0192 \tag{9}$$

$$\mathrm{UC}_{osi} = \overline{C_{osi}} \times 0.022$$

Thus, $\mathrm{UC}_{daco}$ and $\mathrm{UC}_{osi}$ were the drug concentrations to be used in the validation experiments (Supplementary Fig. S11A).

**Phase 1 study design**. All patients included in the phase 1 clinical trial had biopsy-proven metastatic non-small cell lung cancer with somatic activating mutations in EGFR and no prior EGFR inhibitor treatment. Full inclusion/exclusion criteria and study design are provided in the study protocol (Supplement). All patients provided written informed consent and the study was approved by the institutional review board at Memorial Sloan Kettering Cancer Center.

We utilized a standard 3 + 3 dose-escalation design to determine the MTD of dacomitinib and osimertinib and MTD was defined as the highest dose where ≤1 of 6 patients developed dose-limiting toxicity (DLT). DLTs are defined in the protocol and occurred within one cycle (28 days for 1 cycle) after starting treatment with dacomitinib and osimertinib. The dosing levels are provided (Table 3) and an expansion cohort of 10 additional patients treated at the recommended phase 2 dose to further assess the safety and explore preliminary efficacy was planned. Toxicity was graded according to the National Cancer Institute Common Terminology Criteria for Adverse Events CTCAE (CTCAE version 5.0) version 5 and DLTs included are provided in the protocol (Supplement). Toxicity was assessed using descriptive statistics.

**Reporting summary**. Further information on research design is available in the Nature Research Reporting Summary linked to this article.

## Data availability
All data are available on GitHub at Michorlab/NSCLC_OsimDacoOptimization. The clinical protocol is available for review in the Supplementary. Source data are provided with this paper.

## Code availability

Code is available in https://github.com/Michorlab/NSCLC_OsimDacoOptimization.

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

## Acknowledgements

We would like to thank the Michor lab and Daniel Costa for helpful discussions. We gratefully acknowledge support from the Dana Farber Cancer Institute's Physical Sciences Oncology Center (U54CA193461 to F.M.) and the National Institutes of Health (K08CA197389 to A.N.H.). The clinical trial was funded by Pfizer.

## Author contributions

K.E.P., H.A.Y., and F.M. designed the study and wrote the paper. K.P. performed all modeling analyses. F.M. supervised the study. H.Y. was the PI of the clinical trial. A.J.S., A.M., and Y.T. contributed to the clinical trial, Y.W., H.C.-F., M.S., C.N., W.T., A.H., and S.L.W. contributed cell line data, and S.C. and T.O.M. provided input into modeling analyses. All authors edited the manuscript.

## Competing interests

Y.W., W.T., M.S., and S.L.W. are employees of Pfizer and own stock in Pfizer. A.H. has received research funding from Pfizer, Novartis, Amgen, Roche/Genentech, Blueprint Medicines, Relay Therapeutics, and Eli Lilly. H.A.Y. has received research funding to her institution from Pfizer, Novartis, AstraZeneca, Cullinan, Lilly, Daichi. The remaining authors declare no competing interests. The scope of contributions by the Pfizer authors was specifically on preclinical components and did not involve clinical aspects of the work.
