## [Peer Review File · Nature Communications]

Reviewers' Comments:

Reviewer #1:

Remarks to the Author:

The authors present an interesting study in the optimal treatment regimens for combination dosing schedules of Osimertinib (an EGFR inhibitor) and Dacomitinib (a pan-HER inhibitor). The mathematical modeling used in the study combines a birth-death-mutation model of tumor drug-sensitive/resistant clones with pop pharmacokinetic models (with patient-level variability). The approach is suitable for the task, and the methods are extensive, and in general well-presented. However, there are sever major criticisms, and a few minor ones, that need to be addressed.

Major criticisms:

1. The foremost concern is how the proposed schedule is chosen. Figure 3C shows the conventional schedule and proposed schedule – how was this schedule chosen?

The main text hints that other schedules were considered: "Our modeling ensemble identified superior dose-escalation dose levels." Were other schedules run through the modeling ensemble? The authors make the bold claim of "optimal" dosing schedules but only consider 1 schedule!

Secondly, some discussion/analysis of alternative schedules for patients who fail the proposed scheduling treatment would be appropriate. Is this "one-size-fits-all" treatment appropriate for all patients? If not when it fails why does it fail?

2. Turnover/death rates have proven to be important for rates of evolution, clonal expansion, and resistance. The authors have included a supplementary figure on "assumed death rates" in simulations, but I would like further clarification. If I'm correct, the negative growth rate of maximal drug concentration is said to be the death rate (constant for all drug concentrations). Here are maximal drug concentration, birth is assumed to be zero? All other birth rates are calculated as net growth minus death rate (see e.g. figure 1D). Is this correct?

If so, can the birth/death ratio at max drug concentration might be quite different (resulting in same net growth). How would this affect rates of evolution in the birth-death-mutation model?

3. figure 3 – the proposed schedules clearly show percent improvement over conventional. Do these depend on the time interval considered? Same question for figure S10, how do resistance levels compare over varied time intervals?

4. Related to the previous point: in general, the key result of the manuscript is that "higher but less frequent doses of dacomitinib and lower but more frequent doses of osimertinib would yield superior results." It would benefit the manuscript to have an expanded discussion around why this combination works, and what insight is gained by examining the mathematical model.

The introduction only briefly mentions that combination therapy "can delay the emergence of acquired resistance in EGFR-mutant lung cancers," but the logic isn't quite clear until the Methods section.

"If C797S develops in cells without T790M, the cells are resistant to third-generation TKIs, but retain sensitivity to first-generation TKIs. Our tumor evolution assumptions model the last type of resistant mechanism (C797S without T790M)."

Is the "RI" metric of relative improvement of schedule A vs B the best metric for measuring delayed acquired resistance under combination therapy? Relative improvement of resistant subclones, as opposed to relative improvement of overall tumor size?

Minor criticisms:

1. Why isn't C797S shown in figure 1C?

2. Why is birth rate time-dependent in figure 1b – isn't it only drug dependent?

3. It would be helpful if the boxes in fig 2a were colored in corresponding colors to fig b,c. Is the gut compartment not shown?
4. please define QD, BID, TID at first use.

Reviewer #2:

Remarks to the Author:

In this manuscript, the authors address the problem of EGFR targeted therapy resistance in EGFR mutant lung cancer. They develop a multi-variate computational model to identify optimal combination therapy dosing for dacomitinib and osimertinib. The model was tested in vitro, providing some validation of the best combination to thwart drug resistance. A phase I clinical trial was designed and implemented based on the findings, with immature endpoint data.

The problem is important. The approach is interesting and multi-dimensional and tries to address the key issue of tumor heterogeneity and heterogeneity in PK among patients. There are several issues that diminish enthusiasm and to address to improve the depth and impact of the manuscript.

(1) The molecular phenotypes of the cell line models are incomplete. Does the improved "tumor" control in vitro correlate significantly with increased cell death and downstream pathway suppression?

(2) How drug specific are findings? Does the model and validation hold for other inhibitors of the same classes?

(3) As a control, the authors should show that an EGFR mutant cell line model that harbors both T790M and C797S is resistant to the combination therapy across all dose levels.

(4) There are no studies performed in vivo, in particular using a range of EGFR mutant lung cancer PDX models which capture more of the genetic heterogeneity observed in patient tumors.

(5) The response rate in the clinical trial appears low. Response rates to osimertinib alone in these patients are typically higher. How do the authors explain this, especially since the main point is that the modeling aims to identify the optimal dosing regime for phase I dosing. This issue appears to contradict the main thrust of the work.

(6) How does the authors modeling move beyond other related published work that should be discussed in the text for context (e.g. PMIDs: 25349424, 28287179)?

REVIEWER COMMENTS

We would like to thank both reviewers as well as the editor for their careful reading of our paper and their thoughtful suggestions and comments. We have now addressed all comments in a revised version of the manuscript. Please find below a detailed point-by-point response to all issues raised, in which our responses are displayed in blue and edits made to the manuscript are in red.

Reviewer #1 (Remarks to the Author):

The authors present an interesting study in the optimal treatment regimens for combination dosing schedules of Osimertinib (an EGFR inhibitor) and Dacomitinib (a pan-HER inhibitor). The mathematical modeling used in the study combines a birth-death-mutation model of tumor drug-sensitive/resistant clones with pop pharmacokinetic models (with patient-level variability). The approach is suitable for the task, and the methods are extensive, and in general well-presented. However, there are sever major criticisms, and a few minor ones, that need to be addressed.

Major criticisms:

1. The foremost concern is how the proposed schedule is chosen. Figure 3C shows the conventional schedule and proposed schedule – how was this schedule chosen?

The main text hints that other schedules were considered: “Our modeling ensemble identified superior dose-escalation dose levels.” Were other schedules run through the modeling ensemble? The authors make the bold claim of “optimal” dosing schedules but only consider 1 schedule!

Response: Thank you for alerting us to this unclear writing. Indeed, we did consider many other schedules. The goal of the optimization was to identify a schedule that would minimize the number of tumor cells one year after the start of treatment while adhering to optimization constraints such as maximum tolerability previously established with clinicians and commercially available doses in tablets. We searched through multiple schedules to arrive at the optimum conditional on the constraints, which is shown in the main text. As an example, Fig 3D shows the different schedules we considered for level 2 of the dose escalation schedule. We have now revised the manuscript to clarify this point.

Revisions: page 5, line 260, added the following sentence to clarify and summarize our methods: *“All considered schedules adhered to optimization constraints such as tolerability at each dose level and commercially available doses in tablets. Among the considered schedules, we identified the one that would minimize the number of tumor cells one year after the start of treatment.”*

Page 13, line 802, added the following material to describe constraints of considered schedules: *“We considered schedules that adhered to two optimization constraints: 1) an approximation of maximum drug exposure was established with clinicians at each dose level, and 2) schedule options depended on commercial dose availability. Osimertinib is available in 40 mg and 80 mg tablets, whereas dacomitinib is available in 15, 30, and 45 mg tablets. Thus,*

we used these doses in our simulations and varied the frequency of administration while complying to first constraint.”

Secondly, some discussion/analysis of alternative schedules for patients who fail the proposed scheduling treatment would be appropriate. Is this “one-size-fits-all” treatment appropriate for all patients? If not when it fails why does it fail?

Response: Thank you for bringing up this issue. After 8 weeks of treatment, about 80% of simulated patients had a similar outcome with the proposed and conventional schedules (Fig 3e). After one year of treatment, all of 100% simulated patients seemed to perform better with our proposed schedule (Fig 3f). Figure S10B shows a similar trend for patients with a larger BMI. Thus, these findings were summarized as a *one-size-fits-all* approach in the original version of the manuscript. However, to address the comment from this reviewer, we have now expanded the discussion in the paper to clarify the issue. Furthermore, we have also included a discussion on how our approach could be further personalized to individual patients in the future, for instance by obtaining patient-specific PK information before initiating a trial.

Revisions: Page 6, line 283, we added: *“Thus our proposed dose-escalation schedule outperformed the conventional schedule across all simulated patients, even after incorporating inter-patient variability in drug concentration and tumor heterogeneity. This result is encouraging for clinical implementation because customization of schedules for each individual patient was able to be avoided.”*

Page 9, line 444, we added: *“Our modeling platform can be used to personalize treatment for individual patients. We envision that, with the advent of methodology to obtain real-time patient-specific PK information, our models can be used to identify the best dosing schedule for each patient. Drugs that are administered intravenously or drugs that can be obtained in a larger number of concentrations per tablet may be better suited for individual optimization.”*

2. Turnover/death rates have proven to be important for rates of evolution, clonal expansion, and resistance. The authors have included a supplementary figure on “assumed death rates” in simulations, but I would like further clarification. If I’m correct, the negative growth rate of maximal drug concentration is said to the death rate (constant for all drug concentrations). Here are maximal drug concentration, birth is assumed to be zero? All other birth rates are calculated as net growth minus death rate (see e.g. figure 1D). Is this correct?

If so, can the birth/death ratio at max drug concentration might be quite different (resulting in same net growth). How would this affect rates of evolution in the birth-death-mutation model?

Response: Thank you for bringing up this important issue. A negative growth rate of maximal drug concentration from drug assays is indeed due to the death rate. Therefore, at the maximum drug concentration the birth rate is zero, but it is important to note that this maximum concentration was never reached in the simulated clinical trials. Additionally, this assumed death rate was comparable to what has been observed in previous experiments (Fig 5B and D in PMID: 21734175). If we changed the ratio of rates while keeping the net growth rate constant, this would not affect the growth kinetics of one cell type. However, it would affect events generating

a drug-resistant cell from a drug-sensitive cell. In order to investigate this topic, we had performed sensitivity analyses by changing the death rates to those shown in Fig S4b. These death rates are in general lower than the death rates used in our main analysis, shown in Fig S4a, and resulted in a non-zero birth rate at the maximum drug concentration. Nevertheless, we still observed a uniform superiority of our proposed schedule relative to the conventional schedule. The NRAS cell type was an exception, as this cell never displayed a negative growth rate. Thus, we selected a birth rate that would be similar to that of the drug-sensitive cell in the absence of drugs (Fig 2d). In order to clarify these issues, we have revised the methods and results sections.

Revisions: Added a panel to Fig S4 showing simulation results from using time-dependent rates and updated the figure to reflect this change.

3. figure 3 – the proposed schedules clearly show percent improvement over conventional. Do these depend on the time interval considered? Same question for figure S10, how do resistance levels compare over varied time intervals?

Response: Indeed, these results are time-dependent which is why we had considered the results at two different time points in the original version of the paper: after two treatment cycles and 1 year of therapy (Figure 3e-d). We have now rewritten the paper to better explain these results.

Revisions: page 5, line 263 added “Additionally, we opted to compare schedules at two different time-points: after eight weeks of treatment (first planned follow-up of patients) and after one year treatment (when resistance to drug is typically observed).”

4. Related to the previous point: in general, the key result of the manuscript is that “higher but less frequent doses of dacomitinib and lower but more frequent doses of osimertinib would yield superior results.” It would benefit the manuscript to have an expanded discussion around why this combination works, and what insight is gained by examining the mathematical model.

Response: Thank you for this suggestion. We have followed this advice and expanded the discussion directly relating to this key result. In brief, we now describe the drugs’ PK and PD profiles that explain our observations and how our model is able to measure the difference of these schedules in order to arrive this conclusion.

Revisions: Page 8, line 407, added: *“This observation is due to the pharmacodynamic and pharmacokinetic profiles of the individual cell types and drugs under consideration. Dacomitinib has a narrower therapeutic index than osimertinib, and therefore it is necessary to maintain a sufficiently high concentration in plasma, which is reached by administering a larger dose. Also, the drug’s pharmacokinetics suggest a slow depletion in plasma, making frequent doses less necessary to maintain the drug concentration within the therapeutic window (Fig 2B). On the other hand, osimertinib has a large therapeutic index and its concentration in plasma is depleted relatively fast (Fig 2C). As a result, it is not necessary to administer large doses of osimertinib since a lower dose will still reach the therapeutic window, but to maintain the drug-concentration within this window, frequent dosing is required to keep pace with the drug’s rapid elimination from the system. Our mathematical modeling approach yields the additional advantage of a quantitative comparison of the performance of various schedules. Our modeling approach, based on extensive human PK data and in vitro dose response information, provided sufficient support to approve the recommended dosing regimen for a clinical trial, without the necessity of any additional in vivo experiments.”*

The introduction only briefly mentions that combination therapy “can delay the emergence of acquired resistance in EGFR-mutant lung cancers,” but the logic isn’t quite clear until the Methods section.

“If C797S develops in cells without T790M, the cells are resistant to third-generation TKIs, but retain sensitivity to first-generation TKIs. Our tumor evolution assumptions model the last type of resistant mechanism (C797S without T790M).”

Response: Thank you for bringing up this issue. In response, we have rewritten the introduction section to make the logic of this combination therapy more apparent.

Revisions: Page 2, line 60, we added: *“Specifically, osimertinib would be effective in the presence of EGFR T790M while dacomitinib would be effective in the setting of EGFR C797S.”*

Is the “RI” metric of relative improvement of schedule A vs B the best metric for measuring delayed acquired resistance under combination therapy? Relative improvement of resistant subclones, as opposed to relative improvement of overall tumor size?

Response: We have considered other metrics in addition to the RI metric and have found that this metric is less variant with time (as compared to the % change from baseline, PMID: 28566331). We have estimated the relative improvement with regard to the number of resistant clones (Fig S10), but eventually opted to present the relative improvement of the overall tumor cell count since we aimed to identify the schedule that would be most effective across all subclones. We have clarified this issue in the revised version of the paper.

Revisions: Page 13, line 814, we added: *“ $N_S(t)$ is the number of tumor cells of all or one subclone under schedule S. We reported the comparisons of the total cell count to identify schedule that performed better across all subclones.”*

Minor criticisms:

1. Why isn't C797S shown in figure 1C?

Response: Thank you for catching this oversight – we have now added it to the figure and removed it from Figure S2.

Revisions: Updated Figure 1 C to include C797S. Fig1 C is shown below for convenience.

2. Why is birth rate time-dependent in figure 1b – isn't it only drug dependent?

Response: That is correct -- the drug concentration is time-dependent, which is why the birth rate is also time-dependent. We have revised the figure legend to clarify this point.

Revisions: Page 23, figure 1 legend, we clarified by writing: “Each cell type, i , has its own drug-dependent birth rate and constant death rate, $b_i(t)$ and d_i , respectively. Because drug concentration was modeled as a function of time t , the rates are therefore time-dependent.”

3. It would be helpful if the boxes in fig 2a were colored in corresponding colors to fig b,c. Is the gut compartment not shown?

Response: We have implemented these changes. The gut compartment is not shown because the units of drug in this compartment (mg or mol) are different than that of the central and peripheral compartments (nanomolars, nM). In other words, there is no drug concentration in the gut compartment, but a drug amount which is the administered dose.

Revisions: Changed the colors of the central and peripheral compartment in Fig 2a corresponding to Fig 2b. Updated Fig 2 a-c is shown below:

4. please define QD, BID, TID at first use.

Response: Addressed.

Revisions: Page 4, line 201, defined “QD”. Page 5, line 245, defined “BID” and “TID”.

Reviewer #2 (Remarks to the Author):

In this manuscript, the authors address the problem of EGFR targeted therapy resistance in EGFR mutant lung cancer. They develop a multi-variate computational model to identify optimal combination therapy dosing for dacomitinib and osimertinib. The model was tested in vitro, providing some validation of the best combination to thwart drug resistance. A phase I clinical trial was designed and implemented based on the findings, with immature endpoint data.

The problem is important. The approach is interesting and multi-dimensional and tries to address the key issue of tumor heterogeneity and heterogeneity in PK among patients. There are several issues that diminish enthusiasm and to address to improve the depth and impact of the manuscript.

(1) The molecular phenotypes of the cell line models are incomplete. Does the improved "tumor" control in vitro correlate significantly with increased cell death and downstream pathway suppression?

Response: We have included a detailed description of the cell line models including allele frequencies of the driver and resistance mutations in the methods and results sections, as well as provided references to prior characterization of these cell lines. In our prior studies we have observed very close correlation between downstream pathway inhibition and cell viability in response to different classes of EGFR inhibitors that differentially target EGFR resistance mutations (Niederst et al. Clinical Cancer Research 2015, Fig S3C-D. We have now included additional information on independent validation of single-agent dacomitinib and osimertinib

IC50 values in PC9 parental, PC9 T790M, and PC9 C797S cells to provide additional clarity, as well as Fig S3C showing correlation between downstream pathway inhibition and cell viability.

Revisions:

Line 137 added: agreeing with the IC50 value of 51.2 nM (**Fig. S3A**), and consistent with the correlation of viability sensitivity with suppression of downstream signaling (Fig S3C and S3D). Lines 141-144 added: "in line with independent assessment of IC50 values of 11.7 nM and > 1mM, respectively, as well as previous reports comparing sensitivity and suppression of downstream signaling in response to other second and third generation EGFR inhibitors (Niederst et al. CCR 2015)."

Lines 418, 456-473, and 673 added methods and reference to support Fig S3: "*Immunoblotting analysis: Cells were treated with compound for 6 h, harvested, and stored at -80 °C until further analysis. Cell pellets were lysed in lysis buffer (150 mM NaCl, 1.5 mM MgCl₂, 50 mM 4-(2-hydroxyethyl)-1-piperazineethanesulfonic acid (HEPES), 10% glycerol, 1 mM ethylene glycol tetraacetic acid (EGTA), 1% Triton ® X-100, 0.5% NP-40) supplemented with 1 mM Na₃VO₄, 1 mM phenylmethylsulfonyl fluoride (PMSF), 1 mM NaF, 1 mM β-glycerophosphate, cOmplete Mini EDTA-free Protease Inhibitor Cocktail Tablets, and PhosSTOP prior to use. Protein concentration was determined using the BCA Protein Assay (Pierce/Thermo Fisher Scientific, Rockford, IL, USA) per the manufacturer's instructions. Ten μg of total protein were resolved by sodium dodecyl sulfate polyacrylamide gel electrophoresis (SDS-PAGE) and transferred onto nitrocellulose membrane. Blots were probed with primary antibodies overnight at 4 °C in the manufacturer's recommended buffer to detect proteins of interest. After incubation with secondary antibodies, signals were visualized by chemiluminescence (Pierce/Thermo Fisher Scientific) on a FluorChem™ Q digital imager (Protein Simple™, Santa Clara, CA, USA). Antibodies against EGFR (4267), p-EGFR Y1068 (3777), extracellular signal-regulated kinase (ERK) (9102), phosphorylated-ERK (p-ERK) T202/Y204 (9101), and glyceraldehyde-3-phosphate dehydrogenase (GAPDH) (sc-25778) were purchased from Santa Cruz Biotechnology® (Dallas, TX, USA).*"

Added the following figure to the Supplementary data (Fig. S3) with the following description in the legend: "*Dose-response curves of relative cell count to control over drug concentration. IC50 in PC9 cells of dacomitinib and osimertinib are 0.63 nM and 7.51 nM, respectively. The IC50 of osimertinib in the PC9-T790M cell line is 51.2 nM, and IC50 of dacomitinib in PC9-C797S cells is 11.7 nM. (c-d) Signaling pathway inhibition compared with cell viability effects in response to different classes of EGFR inhibitors. (c) Immunoblot analysis of EGFR signaling in PC9 and PC9-DRH cell lines. Cells were treated with the indicated concentrations of dacomitinib (PF-804) or PF-06459988 (PF-9988) for 6h, cell lysates were prepared and analyzed by immunoblotting with antibodies to the indicated proteins and phospho-proteins. PF-06459988 is a third generation EGFR TKI with selective and irreversible activity against EGFR harboring activating mutations (del exon 19 and L858R) as well as T790M, an activity profile closely matched to that of osimertinib¹⁹. (d) Cell viability assay IC50 values for dacomitinib and PF-06459988 in PC9 and PC9-DRH cell lines after 72h of drug treatment.*"

(2) How drug specific are findings? Does the model and validation hold for other inhibitors of the same classes?

Response: From a clinical perspective, the other second-generation EGFR inhibitor (afatinib) is seen as equivalent to dacomitinib and osimertinib is the only approved third-generation EGFR TKI. We conclude equivalence by both comparable efficacy (response rate, progression-free survival) and similar mechanisms of resistance (for second-generation inhibitors resistance is dominated by EGFR T790M).

From a modeling and validation perspective, our approach can be used to design optimum dosing strategies for other combinations as well if data was available to parameterize the model. In order to focus the analysis on the osimertinib/dacomitinib combination as well as discuss the resulting clinical trial, we feel that applying our approach to another combination would not add to the current manuscript. However, we are working on other combinations for future submissions. We have now added a section to the discussion to clarify this point.

Revisions: Line 433, added *“Our mathematical model can be applied to other targeted therapies using proper data to parametrize tumor drug response. For instance, other second-generation*

EGFR inhibitors elicit similar responses to dacomitinib, and we therefore expect that our modeling platform, together with appropriate dose response and PK data, can be applied for designing combination trials for other agents as well.”

(3) As a control, the authors should show that an EGFR mutant cell line model that harbors both T790M and C797S is resistant to the combination therapy across all dose levels.

Response: Thank you for this important comment. We have now performed the appropriate experiments to address this concern. We measured drug-response in a T790M and C797S clone to demonstrate that, as expected, this cell line is resistant to both dacomitinib and osimertinib.

Revisions: Figure S2, we added the following panel that demonstrates the resistance of the T790M and C797S clone. We have also added to the figure legend: “*Combined dacomitinib plus osimertinib does not suppress cells harboring compound T790M and C797S mutations in the used cell line. MGH 121 Res#1 cells with acquired in cis T790M/C797S resistance mutations 38 were treated with increasing concentrations of dacomitinib or osimertinib alone or in the presence of a fixed concentration (1mM) of osimertinib or dacomitinib, respectively. After 72 hours, cell proliferation was assessed by CellTiter-Glo assay. Data are combined from three independent biological replicates (mean, S.E.M.).*”

(4) There are no studies performed in vivo, in particular using a range of EGFR mutant lung cancer PDX models which capture more of the genetic heterogeneity observed in patient tumors.

Response: This is correct. We decided that in vitro and mathematical modeling evidence is sufficient to initiate a clinical trial. This approach is similar to what we have taken in a previous study (PMIDs: 21734175 and 28073786). In addition, there are other ongoing studies of combination EGFR tyrosine kinase inhibitors (osimertinib/gefitinib NCT03122717, EGF816/gefitinib NCT03292133) suggesting that our approach should be tolerable and efficacious. We have revised the manuscript to discuss this issue.

Revisions: Line 418, we added “*Our modeling approach, based on extensive human PK data and in vitro dose response information, provided sufficient support to approve the recommended dosing regimen for a clinical trial, without the necessity of any additional in vivo experiments.*”

(5) The response rate in the clinical trial appears low. Response rates to osimertinib alone in

these patients are typically higher. How do the authors explain this, especially since the main point is that the modeling aims to identify the optimal dosing regimen for phase I dosing. This issue appears to contradict the main thrust of the work.

Response: The aim of combination EGFR treatment is to prevent on-target resistance by inhibiting emergence of EGFR second site mutations. Preventing or delaying resistance would not improve upon the overall response rate but may lengthen the progression-free survival. The response rate to osimertinib single-agent is 80%, which is similar to the 73% we observed in our combination study. However, follow-up is ongoing and we will be able to report on the final efficacy in about 2 years. We have revised the manuscript to address this point.

Revisions: Line 389-395, we have addressed our point above and updated response data: *“Efficacy endpoints remain immature. The median follow-up is 9.7 months. Of the 22 patients treated, 21 have had radiographic assessments of their disease. 16 of 22 have had a confirmed partial or complete response to treatment resulting in an overall response rate of 73%. One patient came off treatment for disease progression. He had concurrent EGFR/TP53/RB1 mutant lung cancer; he did not respond to further treatment and died within 8 months of the diagnosis of metastatic disease. No additional patients have had disease progression. Follow up is ongoing and final efficacy will be reported in the near future.”*

(6) How does the authors modeling move beyond other related published work that should be discussed in the text for context (e.g. PMIDs: 25349424, 28287179)?

Response: Thank you for this comment. Briefly, our work uses a similar stochastic framework as that shown in PMID 25349424, but we allow for time-dependent rates, which are parametrized using in vitro cell line assays. PMID 28287179 uses a different optimization approach, which is deterministic. Neither previous article accounts for the high variability in pharmacokinetics in the clinical setting. We have now expanded the introduction section to address this point and cited these interesting articles.

Revisions: We have cited both of these papers in the introduction in line 67: *“Most approaches have adopted a tumor evolution model with resistant cell clones, yet few account for the complex drug kinetics and the inter-patient variability of drug concentrations (PMIDs: 22982659, 25349424, 28287179).”*

Reviewers' Comments:

Reviewer #1:

Remarks to the Author:

Most of the major criticisms I have previously outlined are addressed in this revised manuscript, however, some points need clarification. The authors should be commended for a very nice example of clinically-relevant research driven by mathematical analysis.

Brief comments on response to major criticisms:

1. One size fits all.

In the revised version the authors note that "customization of schedules for each individual patient was able to be avoided," which is a major advantage to clinical translation. This seems to be a result of the PK model predicting a large dose w/ slow depletion (dacomitinib) or PK model predicting frequent doses w/ fast depletion (osimertinib).

As the authors note in the discussion, personalization of PK may improve patient-specific outcomes, but likely to be similar treatment strategies based on the nature of the therapeutic index of each drugs. Figure 3e,f are powerful, as they show few (if any?) patients which perform at a negative relative improvement with the given metric. Overall tumor size is always improved, even if it comes at the (typically minor) expense of resistant subclones (S10).

However, it is difficult to assess any underlying trend in the waterfall plots to gain any intuition about what patient-specific parameters would be at the largest risk of accelerating resistant subclones in S10. I still would like to understand better the scenarios that accelerate resistance - its an important point and one that deserves a little more attention than is currently given. Even though the proposed treatment schedules seem to have far-reaching optimality in reducing tumor size, some discussion about worst-case scenarios would be very beneficial, and which patient-specific parameters are most important for resistance (figure S10).

2. Turnover / death rates:

Thank you for clarifying how the death rate is chosen, the method employed by the authors seems reasonable here. Further, authors have now shown that major conclusions draw from a constant death rate are not overturned with a new assumption: death as a function of drug concentration. This is a valuable addition to the manuscript.

3. Time-dependent results

This is now clarified, thank you.

4. Benefit of math model

This is also now clarified (see point 2, above). This new expanded discussion makes the utility of the math model clear, and clarifies understanding & aids intuition of the key results presented. Thank you.

Minor criticisms:

I believe that birth rate is a function of the dose concentration within the central compartment (based on Methods, page 12, line 675). I would suggest that the authors denote variables for the compartments in figure 2a (say, $g(t)$, $c(t)$, and $p(t)$), and then clarify that the birth rates in figure 1b are functions of $c(t)$: $b(c(t))$. This is relatively minor point, but an important one if we want to help clarify the model.

Reviewer #2:

Remarks to the Author:

The authors have submitted a revised manuscript. My comments were only partially addressed and there are inaccuracies in the authors' response. First, comments 1,2 and 4 were not adequately experimentally addressed. No cell fate (death, cell cycle arrest, etc.) are shown and no in vivo studies were performed to show that the regimens extend the PFS, as the authors claim they should. These are major gaps that limit the depth of the current study. Second, the authors are incorrect that the cited publications on computational frameworks for establishing drug combinations to combat drug resistance do not take into account PK and variability in drug exposures, in particular for one of the cited publications. Thus, it remains how differentiated and novel the authors' computational framework is in reality. In light of these continued gaps, the manuscript remains a work in progress.

Reviewer #1 (Remarks to the Author):

Most of the major criticisms I have previously outlined are addressed in this revised manuscript, however, some points need clarification. The authors should be commended for a very nice example of clinically-relevant research driven by mathematical analysis.

Response: We would like to thank the reviewer for his/her careful reading of the manuscript and the constructive comments and suggestions. We believe that our paper has become much stronger due to this revision. We would also like to thank him/her for the kind words regarding our work.

Brief comments on response to major criticisms:

1. One size fits all.

In the revised version the authors note that “customization of schedules for each individual patient was able to be avoided,” which is a major advantage to clinical translation. This seems to be a result of the PK model predicting a large dose w/ slow depletion (dacomitinib) or PK model predicting frequent doses w/ fast depletion (osimertinib).

As the authors note in the discussion, personalization of PK may improve patient-specific outcomes, but likely to be similar treatment strategies based on the nature of the therapeutic index of each drugs. Figure 3e,f are powerful, as they show few (if any?) patients which perform at a negative relative improvement with the given metric. Overall tumor size is always improved, even if it comes at the (typically minor) expense of resistant subclones (S10).

However, it is difficult to assess any underlying trend in the waterfall plots to gain any intuition about what patient-specific parameters would be at the largest risk of accelerating resistant subclones in S10. I still would like to understand better the scenarios that accelerate resistance – it’s an important point and one that deserves a little more attention than is currently given. Even though the proposed treatment schedules seem to have far-reaching optimality in reducing tumor size, some discussion about worst-case scenarios would be very beneficial, and which patient-specific parameters are most important for resistance (figure S10).

Response: Thank you for this helpful suggestion. The random forest analysis originally included as Fig S11a-b shows the importance of pre-existing clones and individual PK parameters for predicting the performance of particular schedules after two treatment cycles or one year. For longer treatment, the frequency of pre-existing T790M-positive cells, pT790M, was more important than the proportion of NRAS-mutant cells, pNRAS, for predicting the performance of the schedules. Those patients who had pre-existing T790M-positive clones performed significantly better under our proposed schedule than those without pre-existing clones; the latter still performed better under our schedule but by a smaller margin. In response to the reviewer’s comment regarding the dynamics of resistant subclones during treatment, we have now performed additional analyses. We found that some osimertinib PK parameters correlate with T790M cell numbers during therapy, while dacomitinib PK parameters (absorption rate, clearance of central compartment, and distribution rate) are associated with C797S numbers during therapy. We have now updated the manuscript to address these revisions.

Action taken:

- We updated the paper on line 274 with the following text: “We further explored the relationship of PK parameters with resistant subclones and observed that the clearance of the central and peripheral compartments in the osimertinib popPK model were positively correlated with T790M+ subclones (Fig. S11c, $P < 0.001$, Spearman test with Bonferroni correction). With respect to C797S+ subclones, the clearance of both compartments in the dacomitinib model were positively associated with the subclone count, whereas the absorption rate of dacomitinib was negatively correlated with C797S cell counts (Fig. S11d).”
- Added Figure S11 c-d to the supplementary materials:

- Added the following information to the figure legend: “Random forest analysis to identify important variables. (a) Node purity of variables from random forest. The higher the purity, the more important the variable is when predicting improvement in outcomes. Results are shown for simulations administering two weeks of treatment. (b) Node purity of variables after a year of treatment. The pre-existence and prevalence of NRAS is no longer as important after a year of treatment than after two weeks. All

random forests were run using 50 simulations, with 500 trees in each simulation. Mean node purity with standard errors are shown as black dots and error bars ($n = 50$ estimates), respectively. (c) Scatterplots of T790M+ cell numbers against values of popPK parameters, which are constant across different doses and time. Drug clearance from both compartments as specified in the osimertinib pop-PK model was highly correlated with predicted T790M+ cell count (Spearman correlation test with Bonferroni correction). (d) Scatterplots of C797S+ cells against the popPK parameters. As expected, some PK parameters from the dacomitinib popPK model were significantly correlated with clone size. Parameter descriptions are included in **Table S3.**"

- Added the following supplementary table:

Table S3. Description of popPK parameters from Fig S11.

Parameter	Description
Cl_{co} , CL1o	Clearance of central compartment (osimertinib)
Cl_{po} , CL2o	Clearance of peripheral compartment (osimertinib)
Cl_{cd} , CLd	Clearance of central compartment (dacomitinib)
k_{ad} , kad	Absorption to central compartment (dacomitinib)
k_{ao} , kao	Absorption to central compartment (osimertinib)
Cl_{pd} , Qd	Clearance of peripheral compartment (dacomitinib)
V_{co} , V1o	Volume distribution of central compartment (osimertinib)
V_{cd} , V2d	Volume distribution of central compartment (dacomitinib)
V_{po} , V2o	Volume distribution of peripheral compartment (osimertinib)
V_{pd} , V3d	Volume distribution of peripheral compartment (dacomitinib)

2. Turnover / death rates:

Thank you for clarifying how the death rate is chosen, the method employed by the authors seems reasonable here. Further, authors have now shown that major conclusions drawn from a constant death rate are not overturned with a new assumption: death as a function of drug concentration. This is a valuable addition to the manuscript.

Response: Thank you for suggesting this addition. We are glad that it has clarified our methods.

3. Time-dependent results

This is now clarified, thank you.

Response: Thank you.

4. Benefit of math model

This is also now clarified (see point 2, above). This new expanded discussion makes the utility of the math model clear, and clarifies understanding & aids intuition of the key results presented. Thank you.

Response: Thank you.

Minor criticisms:

I believe that birth rate is a function of the dose concentration within the central compartment (based on Methods, page 12, line 675). I would suggest that the authors denote variables for the compartments in figure 2a (say, $g(t)$, $c(t)$, and $p(t)$), and then clarify that the birth rates in figure 1b are functions of $c(t)$: $b(c(t))$. This is relatively minor point, but an important one if we want to help clarify the model.

Response: Thank you for this suggestion. We have now updated the figure to clarify this point.

Action taken: We have now updated Fig 1b to show that cell division is a function of the concentration, which in turn is a function of time.

We have also added information to the figure legend to clarify each function: “Each resistance mechanism arises in a one-step process. Each cell type, i , has its own drug-dependent birth rate and constant death rate, $b_i(C(t))$ and d_i , respectively. The drug concentrations of dacomitinib, $C_D(t)$, and osimertinib, $C_O(t)$, were modeled as a function of time t . The vector of two drug-concentrations, $C(t) = [C_D(t), C_O(t)]$, serves as the input for the multivariate birth function, b_i . Under a particular drug-dosing schedule, the rates are therefore time-dependent.”

Added labels to Fig 2 b-c to differentiate the functions of the concentrations between the central and peripheral compartments. These new functions are referenced in the legend of Fig 1b. Changed Fig 2b-c legend to: “(b) and 40 mg QD of osimertinib (c) in 1000 patients. Blue and red lines correspond to drug concentration in the central ($C_D(t)$ and $C_O(t)$), and peripheral, ($P_D(t)$ and $P_D(t)$), compartment, respectively. Solid lines are median concentrations and shaded areas represent a 95% confidence interval.”

Reviewer #2 (Remarks to the Author):

The authors have submitted a revised manuscript. My comments were only partially addressed and there are inaccuracies in the authors' response. First, comments 1,2 and 4 were not adequately experimentally addressed. No cell fate (death, cell cycle arrest, etc.) are shown and no *in vivo* studies were performed to show that the regimens extend the PFS, as the authors claim they should. These are major gaps that limit the depth of the current study. Second, the authors are incorrect that the cited publications on computational frameworks for establishing drug combinations to combat drug resistance do not take into account PK and variability in drug exposures, in particular for one of the cited publications. Thus, it remains how differentiated and novel the authors' computational framework is in reality. In light of these continued gaps, the manuscript remains a work in progress.

Response: We very much appreciate that the reviewer took the time to readdress the points made in the first round of revisions. We apologize for any omissions, misunderstandings and inconsistencies that might have arisen/ remained, which have now been fully addressed. Below we organize our response by separating the comments from the first revisions and devoting each point to a response to each comment from reviewer #2.

Comment 1: *"The molecular phenotypes of the cell line models are incomplete. Does the improved "tumor" control in vitro correlate significantly with increased cell death and downstream pathway suppression?"*

Response: Despite not parametrizing death rates from apoptosis experiments in the original version of the manuscript, we did perform a sensitivity analysis by varying the death rates of the cells according to drug concentration (Fig S4 and see response to reviewer #1 comment #2). We found that our recommendations were robust to these changes in death rate parametrization. In response to the reviewer's comment, we have now additionally performed *in vitro* experiments to characterize the

molecular phenotypes of PC9 (Del) and RPC9-CL6 (Del/T790M) with EGFR inhibitor treatment. As shown in the new Figure S3C, the combination treatments with the five combination conditions (as specified in Figure 3C and Figure 4) resulted in stronger suppression of EGFR and downstream cell signaling and induction of apoptosis than osimertinib or dacomitinib alone in PC9 (Del) or RPC9-CL6 (Del/T790M) cells, respectively. We have updated the main text and supplement to reflect these new experimental findings.

Action taken:

- Added the following material to the main text:

Line :“PC9-DRH cells, which harbor both the single-mutant (exon 19 del) and double-mutant (exon 19 del/T790M) alleles, were found to be resistant to dacomitinib even at 250 nM, but responded to osimertinib at 37 nM and higher concentrations (-0.007 log-cells h⁻¹ decrease in slope, p = 0.017), agreeing with the IC50 value of 51.2 nM (Fig. S3a), and consistent with the correlation of viability sensitivity with suppression of downstream signaling and induction of apoptosis (Fig. S3c).”

Line 314: “The suppression of EGFR pathway and induction of apoptosis induction in PC9 and RPC9-CL6 are consistent with long term viability effect of the combination (Fig S3C).”

- Added procedures of immunoblot assays to Methods (starting on line 466).
- Added the following to the supplementary material:

Fig S3c legend: “Immunoblot analysis of EGFR signaling in PC9 and RPC9-CL6(Del/T790M) cell lines. Cells were treated with the indicated concentrations of dacomitinib (Daco) or osimertinib (Osi) for 24h, cell lysates were prepared and analyzed by immunoblotting with antibodies to the indicated proteins and phospho-proteins.”

Comment 2: *“How drug specific are findings? Does the model and validation hold for other inhibitors of the same classes?”*

Response: We did not experimentally address this question because in this paper, we were interested in the combination of osimertinib and dacomitinib to identify an optimal dosing schedule for the phase I/II trial we subsequently conducted at MSKCC. Currently, we are constructing similar models to identify optimal schedules using other inhibitors of the same class in combination with different class inhibitors for the treatment of EGFR-m NSCLC, but these are separate projects. We have updated the text to reflect this approach.

Action taken: Added the following to the discussion, line 444:

“We intend to increase the practicality of our model by incorporating other resistance mechanisms commonly observed in treated EGFR-m NSCLC and feature other targeted therapies in combination with EGFR inhibitors and will do so in future applications.”

Comment 4: *“There are no studies performed in vivo, in particular using a range of EGFR mutant lung cancer PDX models which capture more of the genetic heterogeneity observed in patient tumors.”*

Response: Thank you for raising this issue. We did not experimentally address this comment because, when starting this project, we had established a short timeline for a phase I trial. Since each drug has been studied rigorously as a single treatment and was proven to be tolerable and effective, and the proposed trial protocol was accepted by the MSKCC IRB, we opted to not do *in vivo* experiments. We have now commented on this choice in the revised version of the text.

Action taken: Added the following text to the discussion section, line 367:

“Our modeling approach, based on extensive human PK data and in vitro dose response information, provided sufficient support to approve the recommended dosing regimen for a clinical trial, without the necessity of any additional in vivo experiments. We therefore performed a phase 1b study incorporating the dosing schedule derived from our predictive modeling platform and demonstrated the safety and tolerability of the combination, a proof of concept that could justify broader use of this technique in future phase 1 combination trials.”

Lastly, we would like to respond the reviewer’s comment about an inaccuracy from our response, particularly regarding our previous paper (PMID 22982659 Foo et al 2012). In that paper, we used a non-compartmental exponential model to describe erlotinib PK. In this model, the only variation of drug concentrations between patients came from the patients’ smoking status, as an example of how PK variability could arise and how large it could be between different patient groups. These PK estimates were based on 32 male subjects in a two-arm study (PMID: 16609030). Unfortunately, at the time at which we wrote the 2012 paper, we were not able to include any other additional PK data. In contrast, in the current paper, we use population PK models obtained from larger patient cohorts; these models include several other predictive factors for PK dynamics. After all variables are accounted for in the model, there is an

additional and unexplained variability in drug-concentration between patients, which is incorporated into our modeling platform. We have now reformulated the corresponding section in the introduction to make this distinction more apparent.

Action taken: Changed the corresponding section in the introduction to the following on page 66: *“Most approaches have adopted a tumor evolution model with resistant cell clones in different patient groups^{13,16,17}, yet few account for the complex pharmacokinetics and the inter-patient variability of drug concentrations identified from large patient cohorts.”*